

# Assessing a low-cost methane sensor quantification system for use in complex rural and urban environments

Ashley Collier-Oxandale[1], Michael P. Hannigan[2], Joanna Gordon Casey[2], Ricardo Piedrahita[3], John Ortega[4], Hannah Halliday[5], & Jill Johnston[6]

[1]Department of Environmental Engineering, University of Colorado Boulder, Boulder, CO, 80309, USA
[2]Department of Mechanical Engineering, University of Colorado Boulder, Boulder, CO, 80309, USA
[3]Berkeley Air Monitoring Group, Boulder, CO, USA
[4]National Center for Atmospheric Research, Boulder, CO, 80301, USA
[5]NASA Langley Research Center, Hampton, VA, 23666, USA
[6]Department of Preventive Medicine, University of Southern California, Los Angeles, CA 90089, USA

*Correspondence to*: Ashley Collier-Oxandale (ashley.collier@colorado.edu)

**Abstract.**

Low-cost sensors have the potential to facilitate the exploration of air quality issues on new temporal and spatial scales. Here we evaluate a low-cost sensor quantification system for methane through its use in two different deployments. The first, a one-month deployment along the Colorado Front Range includes sites near active oil and gas operations in the Denver-Julesburg basin. The second deployment in an urban Los Angeles neighborhood, an subject to complex mixture of air pollution sources including oil operations. Given its role as a potent greenhouse gas, new low-cost methods for detecting and monitoring methane may aid in protecting human and environmental health. In this paper, we assess a number of linear calibration models to convert raw sensor signals into ppm concentration values. We also examine different choices that can be made during calibration and data processing, and explore cross-sensitivities that impact this sensor type. The results illustrate the accuracy of the Figaro TGS 2600 sensor when methane is quantified from raw signals using the techniques described. The results also demonstrate the value of these tools for examining air quality trends and events on small spatial and temporal scales as well as their ability to characterize an area – highlighting their potential to provide preliminary data that can inform more targeted measurements or supplement existing monitoring networks.

# 1 Introduction

## 1.1 Background and Motivation

Given both the direct impacts on climate change and indirect impacts on human health, it is important to study increased atmospheric methane on varied temporal and spatial scales. Methane is an important greenhouse gas with 28 times the global warming potential of $CO_2$ over a 100-year lifetime (IPCC, 2015); moreover, the majority of methane emissions result from human activity (US EPA, 2017). Researchers using ice core samples to measure historic methane levels found relatively stable atmospheric concentrations of approximately 0.695 ppm from 1000 AD until the Industrial Revolution (Etheridge et





al., 1988), after which methane concentrations have grown to a present-day global average of 1.851 ppm (NOAA, 2017). Not only does this increased atmospheric methane intensify climate change, but it also contributes to higher ground-level ozone – a public health risk (Fiore, 2008). Multiple modeling studies have revealed the benefits of reducing methane emissions, which include decreased premature mortality from respiratory illness caused by ozone (West et al., 2006; Fang et al., 2013).

A better understanding of emissions and sources could help in the effort to reduce atmospheric methane.

In 2015, production, storage, processing, and distribution of natural gas and petroleum were responsible for approximately one-third of methane emissions in the US (US EPA, 2017). Vented and fugitive emissions of methane that occur at oil and gas production sites may raise concerns for nearby communities due to potential co-emission of hazardous BTEX (benzene, toluene, ethylbenzene and xylene) compounds (Adgate et al., 2014; Helmig et al., 2014; Moore et al., 2014). While all of the

leaks along the chain of production, processing, transmission, and distribution contribute to climate change. Recent studies also suggest that methane emissions from the oil and gas sector are underestimated in current inventories (Miller et al., 2013, Wilcox et al., 2014; Zavala-araiza et al., 2015; Petron et al., 2014; Subramanian et al., 2015). Miller and co-workers found that methane emissions in US EPA inventories may be underestimated by a factor of 1.5 (Miller et al., 2013). It has been suggested that these discrepancies between measured methane and source-based inventory estimates may be explained by

"super-emitters" – a small percentage of sites or equipment that contribute a large portion of the emissions (Wilcox et al., 2014, Petron et al., 2014). For example, a study in the Barnett Shale region found that at any given time, two percent of facilities accounted for half of methane emissions and that these sites vary spatiotemporally (Zavala-araiza et al., 2015). As described in a recent review, smart-sensing systems designed to detect leaks and alert operators at the well pad level may aid in identifying these events as they occur (Allen, 2014), speaking to the need for tools that can feasibly achieve useful spatial

and temporal resolution for monitoring at the local or facility level.

## 1.2 Low-Cost Sensors for Air Quality Monitoring

### 1.2.1 A Place for Sensors

Typically monitoring methods and technologies are driven by the research question of interest and available resources. For example, the National Oceanic and Atmospheric Administration (NOAA) has maintained a global monitoring network for

methane for upwards of 30 years to study long-term atmospheric trends, seasonal cycles, and its global distribution (NOAA, 2017). Monitoring networks can also be built on smaller scales to study methane fluxes at the regional or city-level; the MegaCities Project is currently undertaking this work in Southern California as is the INFLUX project in Indianapolis (Wong et al., 2015, Davis et al., 2017). Remote sensing provides a global picture and, given the spatial coverage, this data can highlight hotspots at the regional-scale (Kort et al., 2014). However, interferences and satellite trajectories prevent truly

continuous data collection for any single location. Mobile monitoring using vehicles equipped with gas analysers as well as aircraft campaigns both allow for horizontally and vertically resolved spatial coverage at the neighborhood or facility level. Additionally, these methods facilitate the collection of high-quality data with precise instrumentation (Yacovitch et al., 2015;



Karion et al., 2013). However, aircraft data typically represent a "moment in time" and changing meteorological conditions often limit the ability to repeat data collection. Ground-based mobile monitoring may be repeated more easily, however, the data collected is often periodic in nature and intended for targeted studies. Currently the scientific and regulatory communities are limited in their capability to collect data continuously at the neighborhood or facility level. While it would

be possible to site the same high-quality instruments utilized in global and regional monitoring networks at a local scale, this approach would be costly given the expense of the equipment, the siting requirements, and the expertise needed for operation.

Low-cost air quality sensing systems are well suited to fill this role by providing continuous measurements in high-density networks at a local scale. Given their versatility and capacity to provide high-spatial and temporal resolution data these

systems could augment regulatory monitoring systems, aid in compliance monitoring (e.g., leak detection), or enable the public to formulate local strategies to reduce their exposure (Snyder et al., 2013). These systems are relatively easy to deploy and operate in nearly any type of location due to their size, low power requirements, and automated electronic data collection. These characteristics also make them more accessible for community-engaged research applications than conventional methods (Shamasunder, et al., 2017). For example, these systems could support a community collecting

preliminary data, in partnership with researchers or local regulatory agencies, that could be evaluated for "hotspots" or correlated with community members' experiences (e.g., odors or health symptoms) – providing more information to support better understandings of complex air quality issues.

### 1.2.2 Previous Sensor Research

Several studies have demonstrated the ability of low-cost sensors to measure pollutants of interest at ambient levels. For

example, CO, NO, and $NO_2$ have been measured in an urban sensor network with additional analysis demonstrating the ability to differentiate local emissions from regional trends (Mead et al., 2013; Heimann et al., 2015). In another example, researchers demonstrated the feasibility of collecting personal CO, $NO_2$, $O_3$, and $CO_2$ exposure data with uncertainty estimations using a portable, wearable system (Piedrahita et al., 2014). Several studies have also made use of sensors to study the spatial variability of $O_3$ on various scales (Sadighi et al., 2017; Cheadle et al., 2017; Moltchanov et al., 2015).

Connected to this effort on sensor applications, there has been much work evaluating the performance of individual sensors (Masson et al., 2015a, 2015b; Spinelle et al., 2015, 2017; Lewis at al., 2015) and demonstrating the performance of different calibration approaches (Zimmerman et al., 2017; Kim et al., 2017; Cross et al., 2017).

While many projects utilize sensors capable of detecting criteria pollutants, advances in the development of metal-oxide semiconductor (MOx) sensors have led to sensors capable of detecting methane in settings closer to ambient environmental

conditions (Quaranta et al., 1999; Biaggi-Labiosa, 2012). Eugster and Kling (2012) demonstrated the ability of the Figaro TGS 2600 sensor to resolve diurnal methane fluctuations in a remote area of Alaska. A similar sensor, the Figaro TGS 2611-E00, was found to have an accuracy of ±1.7 ppm in a laboratory setting for minute-averaged data – suggesting its suitability for detecting substantial methane leaks (Van den Bossche et al., 2017). These and similar metal-oxide VOC sensors have



also been utilized in other applications such as odor detection at landfills and electronic noses (Penza et al., 2015; Zhang et al., 2008).

This paper describes a methodology for collecting and quantifying data using Figaro TGS 2600 MOx sensors to examine ambient trends and methane enhancements on small spatial and temporal scales. Data from two field deployments is used to

discuss the different considerations for calibrating and deploying these sensors. The first dataset was collected in Colorado during the FRAPPE/DISCOVER-AQ monitoring campaign in the summer of 2014 (Pfister et al., 2017). This deployment primarily measured rural and semirural areas along the Front Range north of Denver; important sources of methane in the area include oil and gas development and agriculture/ranching. The second data set was collected in California near downtown Los Angeles in the late-summer/early-fall of 2016 as part of a community-based research project. This

deployment was in a mainly urban area with high-density housing near two major roadways and urban oil extraction. With this work, we build on the previous study by Eugster and Kling (2012) by demonstrating the use of these sensors in more complex environments where they are likely subject to a greater number and variety of local and regional influences. We (1) demonstrate methods for sensor calibration and validation of the Figaro TGS 2600 MOx sensors using field co-locations, (2) examine different options and issues that arise in the calibration process, and (3) explore the potential for the field data from

these sensors to offer unique information. This paper is intended to explore ways of adapting this system to fit the needs and logistical constraints of different investigations in order to provide useful and relevant methane estimations.

## 2 Methods

### 2.1 Instrumentation – Low-Cost Sensor Systems

In both deployments, embedded sensor systems termed U-Pods and Y-Pods (subsequent iterations of an open source

platform) were used for data acquisition (mobilesensingtechnology.com). The main differences between the two versions were in the circuit board design and the programming, which was altered to improve reliability. Each U-Pod/Y-Pod (Pod) was outfitted with multiple gas-phase and environmental sensors, listed in Table 1. The two Figaro VOC sensors were originally developed for monitoring in industrial applications where much higher pollutant concentrations are expected, compared to ambient environmental monitoring. The following analysis will primarily utilize signals from one of these VOC

sensors – the Figaro TGS 2600 MOx sensor. This is the same sensor used by Eugster and Kling (2012) in Alaska, deployed here in environments characterized by complex mixtures including methane emissions and associated confounding gas species.

**Table 1: U-Pod and Y-Pod Sensor Lists**

| Sensor Type | U-Pod | Y-Pod |
|---|---|---|
| Temp. & RH | RHT03 (aka DHT22) | Sensirion SHT2 |
| Temp. & Pres. | 47 Bosh BMP085 | Bosh BMP180 |
| Carbon dioxide | ELT S-100 NDIR | ELT S-300 NDIR |



| Ozone | SGX Corporation MiCS-2611 | SGX Corporation MiCS-2611 |
|---|---|---|
| VOC Sensor 1 | Figaro TGS 2600 MOx | Figaro TGS 2600 MOx |
| VOC Sensor 2 | Figaro TGS 2600 MOx | Figaro TGS 2600 MOx |
| Additional Optional Sensors | Alphasense B4 series (CO, NO, NO2, O3, SO2), Baseline Mocon PID | Alphasense B4 series (CO, NO, NO2, O3, SO2), Baseline Mocon PID |

These embedded sensor systems are housed in small weather-proof plastic cases (approximately 20 cm x 25 cm x 10 cm) with fans to pull ambient air through the enclosure and across the sensor surfaces resulting in multiple air exchanges occurring each minute. The systems in these weatherproof cases can be placed outdoors for long-periods of time. They are

powered using 12V AC/DC adapters plugged into wall power, but can use car batteries and/or solar power in remote locations. All data is logged to an onboard micro-SD card. As configured, these Pods draw roughly 11 Watts. These systems have been used in several other indoor and outdoor air quality studies (Casey et al., 2017; Sadighi et al., 2017, Cheadle et al., 2017). Figure 1 includes a labeled photo of a Y-Pod interior and a photo of two Y-Pods deployed.

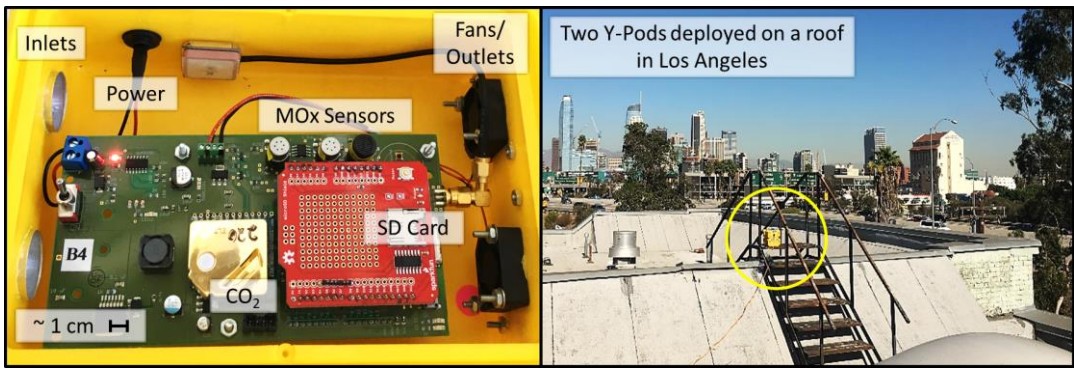

**Figure 1: A labeled photo of a Y-Pod interior (left) and a photo of two Y-Pods deployed at a field site (right).**

**2.2 Deployment Overview**

Regulatory monitors have strict siting guidelines; however, no such constraints exist for low-cost sensor systems. Once a site is selected, Pod placement is chosen based on feasibility, access to air flow, and avoiding potential obstructions as much as possible to obtain samples that are representative of the area. As selecting sampling locations and setting up Pods is typically

a joint effort with community partners, different sites often require different approaches. For example, at multi-story building Pods are typically placed on the roof, while at a single-family home we may place the Pod on the edge of a first story roof or a fence. Additional considerations include access to power, whether the instrument is obstructing a walkway or driveway, and safety of the residents. In both deployments discussed in this paper, site selection was guided by the research goals and access to general air flow while also considering the preferences of the owner, tenants, or site manager.

In Colorado, the Pods were used during the 2014 FRAPPE/DISCOVER-AQ campaign with the aim of characterizing small-scale spatial variability of pollutants. This deployment lasted roughly one month. The deployment of the Pods was centered on a main site for the FRAPPE/DISCOVER-AQ campaign, the Boulder Atmospheric Observatory (BAO) Tower in Erie,



Colorado. Fourteen Pods were placed in an approximately 10x10 km grid. The remaining four Pods were placed to the southwest and northeast of the grid to provide regional comparisons, with measurements taken at Golden/NREL, Frederick, and Platteville. All Colorado sites are shown in Figure 2.

Also, shown in Figure 2 are the boundaries of the Wattenberg Gas Field and active and inactive wells. In general, oil and gas activity increases in density moving from the south-eastern side of the deployment region to the north-western side, with the Erie sites on the edge of the gas field. Note, the Golden/NREL site has no nearby oil and gas activity, while the Platteville site is surrounded by a high density of wells. The Pods were sited in rural/suburban areas primarily at homes, schools, or in open-space with two monitors sited at a water reclamation facility. Of the eighteen monitors, data from fifteen were included in the following analysis. Three monitors were excluded because of extended power failure, temperate/humidity sensor failure, or MOx VOC sensor malfunction. The remaining fifteen monitors provided nearly complete datasets and all necessary sensors operated continuously.

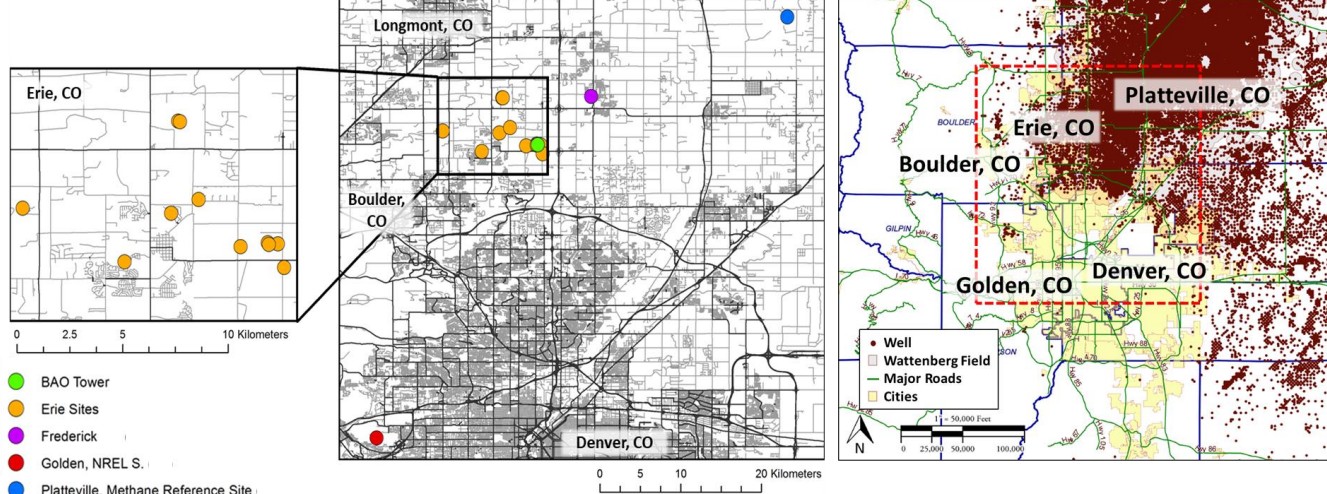

**Figure 2: Maps of Colorado deployment area with every site displayed in the center map and the Erie sites displayed in the map inset on the left. The map to the right indicates active and inactive wells in the Wattenberg Gas Field along with major urban areas and counties, data courtesy of the Colorado Oil and Gas Conservation Commissions (COGCC, 2017).**

In Los Angeles, we partnered with two community-based organizations, Redeemer Community Partnership and Esperanza Community Housing, and deployed Y-Pods throughout a neighborhood south of downtown Los Angeles. This deployment lasted approximately eight weeks. The community was specifically interested in deploying a monitoring network around an active oil extraction site. In this case, sites were selected at varying distances away from the drilling operation as well as varying distances from freeways, another potential source of pollutants (Figure 3). Thirteen of the sites were within an approximately 5x5 km grid, and two additional sites were located further to the northwest and northeast. These two additional sites were utilized because they allowed for continuous co-location with reference instruments for validation purposes. The deployment area in Los Angeles, was primarily urban/suburban with high-density residential areas, some commercial and industrial land use, and much higher density traffic than the Colorado deployment area.



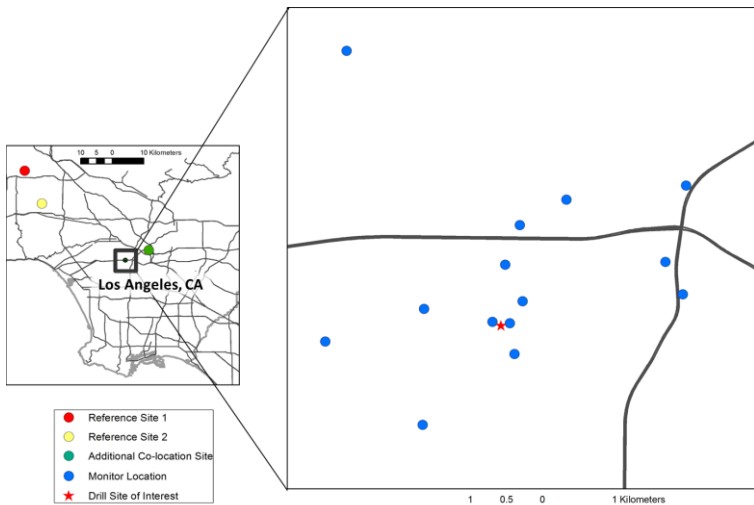

**Figure 3: Map of Los Angeles deployment sites, the map on the left showing the deployment area and all sites where co-locations with reference monitors occurred, the map on the right showing the distribution of monitoring sites in relation to major roadways and the drill site of interest (note the monitor locations have been approximated to the center of their respective blocks to protect participant identities).**

Evaluating the performance of the Figaro TGS 2600 MOx sensor in the context of these two deployments provides the opportunity to better understand its strengths and limitations. In Colorado, the sensor network covered a larger area and we examined methane trends with respect to regional differences in potential sources. In Los Angeles, the sensor network covered a smaller area to examine local methane trends and to attempt to distinguish emissions from point sources. Another important distinction between the two locations is the nature of the oil and gas activity. In Colorado, the deployment was in the SW portion of the Denver-Julesberg Basin, which produces a mix of natural gas, condensate liquids, and crude oil (US EIA, 2016a). This area also includes the Wattenberg Field, which ranked in the top ten for both oil and gas producing fields in 2013 (US EIA, 2015). In Los Angeles, oil and gas activity refers primarily to crude oil production. California is the fourth top producing states for crude oil (US EIA 2016b), and Los Angeles County is home to more than 5,000 active oil wells (Sadd and Shamasunder, 2015). In both cases we expect methane to be emitted or co-emitted with other VOCs and as we attempt to better understand local sources, methane may serve as a valuable indicator of emissions from these types of sites. The ratio of methane relative to other combustion products such as CO and $CO_2$, will likely be higher from sites related to oil and gas activity than from other local sources such as traffic. (Nam et al., 2004; Popa et al., 2014; Pischl et al, 2013). While the two deployment locations offer contrasting sampling environments, both locations offer complexity in terms of number/types of sources, geography, and typical atmospheric trends.

## 2.3 Sensor Signal Processing

The operating principle of MOx semiconductor sensors is based on a reducing gas changing the resistance of a semiconductor material in a simple resistance circuit (Sun et al., 2012). In clean air, the flow of current across the sensor



surface is limited by donor electrons in the tin dioxide that are attracted to oxygen adsorbed to the sensor's surface. The flow of current increases when the target gas (e.g., methane) is present, thus reducing the amount of oxygen adsorbed to the sensor's surface (Figaro, 2005a). In other words, the resistance across the sensor decreases with increasing methane. In both the Y-Pods and U-Pods, the sensor voltage is continuously recorded to the SD card. Using Eq. 1, provided by the sensor
manufacturer, we calculate the sensor resistance ($R_s$) at various concentrations (Figaro, 2005b). In this equation $V_c$ is the circuit voltage, $R_l$ is the load resistance, and $V_{out}$ is the logged voltage. $R_0$ represents the resistance in clean air and the ratio of $R_s/R_0$ is typically used in the analysis of MOx sensor data (Eugster and Kling, 2012; Piedrahita et al., 2014). Gas sensor signals, temperature, humidity, and pressure are recorded to the SD card approximately every 6-25 seconds (depending on a Pod's programming). This frequent data acquisition allows for the use of minute-median data in calibration and analysis.
Unless otherwise stated, this is the time-resolution used in our analysis and shown in this paper.

$$R_S = \frac{V_c * R_L}{V_{out}} - R_L \qquad (1)$$

**2.4 Sensor Calibration, Validation, and Analysis**

Field normalizations were used to generate calibration models for the sensors. Field normalization provides one approach to correcting for the cross-sensitivities low-cost sensors tend to exhibit with respect to temperature, humidity, and other trace
gases (Spinelle et al., 2015, 2017; Sadighi et al., 2017, Wang et al., 2010) and this method is implemented by co-locating low-cost sensor systems with high-quality reference instruments (typically regulatory-grade monitors) for a given period and then generating a calibration model using an approach such as linear regression. These calibration models predict the methane concentration (in ppm) based on the sensor signal ($R_s/R_0$) and other predictors. An advantage of calibrating sensors in the field as opposed to in a laboratory setting is that the models will be trained for the pollutant levels of interest, across
the same dynamic temperature and humidity values that a sensor will likely experience during field deployment. In a study involving personal air quality monitors, Piedrahita and co-workers (2014) successfully calibrated sensors and provided sensor-specific uncertainty estimates using this method.

In Colorado, we co-located U-Pods with a Los Gatos cavity ring-down spectrometer operated by the Penn State Native Trailer team at the Platteville Atmospheric Observatory in Platteville, CO. In Los Angeles, we co-located Y-Pods with
reference instruments at two different sites. The pre-deployment co-location was with a Baseline Mocon Series 900 Methane/Non-methane Hydrocarbon Analyzer located in a primarily residential suburban area of Los Angeles. The post-deployment calibration was with a Picarro cavity ring-down spectrometer located in a suburban/urban area with a mix of residential, retail, and industrial land use. Reference instruments at both Los Angeles sites were operated by the South Coast Air Quality Management District. The timelines in Figures 4 and 5 illustrate when Pods were co-located vs deployed in the
field as well as, which data was used for the generation of calibration models (i.e., training data) versus the validation of those models (i.e., testing data). Note for the Colorado deployment, both before and after the field deployment, the montiors were co-located in batches due to logistical constraints. Arrows indicate the movement of batches of montiors, and the "Not



in Use" row clarifies whether Pods were deployed. In additon, during the Colorado deployment, a single calibration model (a universal model) was developed based on the data from the "Main U-Pod", described in greater detail below. For the Los Angeles deployment, calibration models specific to each Y-Pod (sensor-specific models) were used.

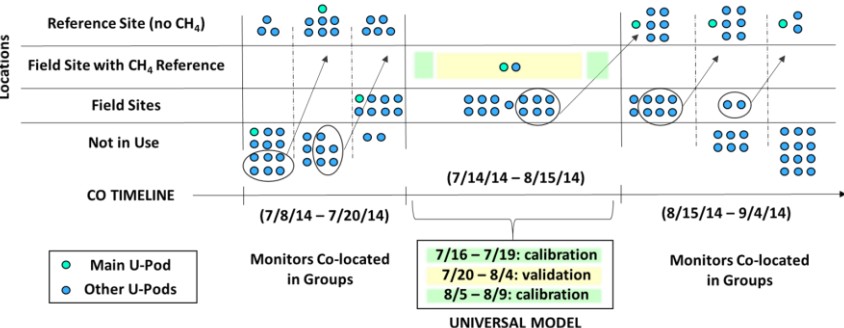

**Figure 4: Timeline for Colorado, indicating when monitors were co-located together in batches before and after the field deployment, as well as, illustrating how 2 U-Pods were sited with a reference instrument during the field deployment (and this data was used for calibration generation or training data verses model validation or testing data)**

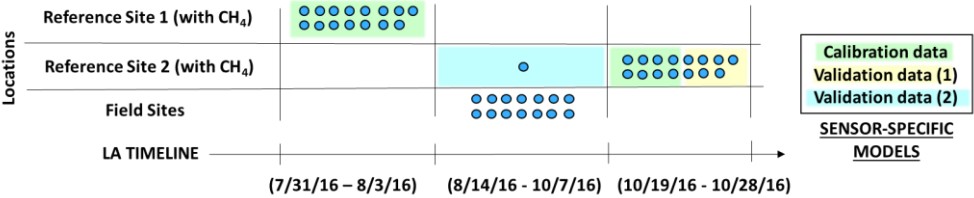

**Figure 5: Timeline for LA indicating when Y-Pods were co-located together with reference instruments before and after the field**
**deployment, as well as which data was used for calibration generation (training data) verses model validation (testing data)**

For both deployments, four days of data at the beginning and four days at the end of the co-location with reference instruments were used to generate the calibration models.  Specifically, for Colorado, four days at the beginning and end of the field deployment were used for generating the quantification model; while in Los Angeles, four days from both the pre and post co-location were used for model generation. The remaining data from co-locations was then used for model

validation (approximately eighteen days for Colorado and four days for LA). Table 1 lists the calibration models that were compared. Several models (the simpler ones) selected are commonly used in sensor calibration, while the more complicated models were selected based on predictors that aided in correcting for cross-sensitivity and resulted in more normal residuals. The models are listed in order of their complexity, beginning with the addition of environmental parameters, then interactions, and then transformations. Regression analysis provides sensor-specific coefficients for predictor variables. The

models are then inverted, so that gas concentration is expressed as a function of sensor signals and can then be used to predict pollutant concentrations using new data collected in the field. This inverted model approach is typical for field normalization (Piedrahita et al., 2014; Spinelle et al., 2015, 2017). Evaluation of model performance was based on the coefficient of determination and the root mean squared error, as well as an analysis of the residuals in relation to relevant





environmental and air quality parameters. Validation data provides the opportunity to evaluate the consistency of each model's performance based on the same metric but with the addition of mean bias.

**Table 2: Calibration Models**

| Mdl # | Description | Model Equation |
|---|---|---|
| 1 | Mdl1 | $R_s/R_0 = p_1 + p_2(C)$ |
| 2 | Mdl3 | $R_s/R_0 = p_1 + p_2(C) + p_3(T) + p_4(H)$ |
| 3 | Mdl4 | $R_s/R_0 = p_1 + p_2(C) + p_3(T) + p_4(H) + p_5(T_i)$ |
| 4 | Mdl4_1Int | $R_s/R_0 = p_1 + p_2(C) + T(p_3 + p_6(C)) + p_4(H) + p_5(T_i)$ |
| 5 | Mdl4_2Int | $R_s/R_0 = p_1 + p_2(C) + T(p_3 + p_6(C)) + p_4(H) + p_5(T_i) + p_7(T * H)$ |
| 6 | Mdl4_1Int_Tr | $R_s/R_0 = p_1 + p_2(C) + T(p_3 + p_6(C)) + p_4(ln(H)) + p_5(T_i)$ |
| 7 | Mdl4_2Int_Tr | $R_s/R_0 = p_1 + p_2(C) + T(p_3 + p_6(C)) + p_4(H^{-1}) + p_5(T_i) + p_7(T * H^{-1})$ |
| 8 | Mdl5_1Int | $R_s/R_0 = p_1 + p_2(C) + T(p_3 + p_6(C)) + p_4(H) + p_5(T_i) + p_7(T_d)$ |
| 9 | Mdl5_2Int | $R_s/R_0 = p_1 + p_2(C) + T(p_3 + p_6(C)) + p_4(H) + p_5(T_i) + p_7(T * H) + p_8(T_d)$ |
| 10 | Mdl5_1Int_Tr | $R_s/R_0 = p_1 + p_2(C) + T(p_3 + p_6(C)) + p_4(ln(H)) + p_5(T_i) + p_7(T_d)$ |
| 11 | Mdl5_2Int_Tr | $R_s/R_0 = p_1 + p_2(C) + T(p_3 + p_6(C)) + p_4(H^{-1}) + p_5(T_i) + p_7(T * H^{-1}) + p_8(T_d)$ |

**Predictors (lower case p with subscripts) are C – pollutant concentration (ppm methane), T – temperature, H – absolute humidity, $T_i$ – continuous time, $T_d$ – categorical time of day, and model identifiers are Mdl# - indicates the number of predictors, #Int – indicates the number of interactions, Tr – indicates use of transformations. The predictor $p_1$ indicates an empirical constant.**

Given the structure of each deployment and availability of co-located data, two different approaches to developing and applying calibration models were used: a universal calibration model vs. sensor-specific models. For the Colorado deployment, a universal calibration model was developed using the data from one sensor and this model was applied to all the sensors. As shown in Figure 4, two U-Pods were co-located with the reference instrument throughout the field deployment. The data from one of these Pods was used along with the following process: (1) generate a universal calibration model using data from the "Main U-Pod" co-located with the reference instrument, (2) normalize all of the other U-Pods' raw sensor signals to the "Main U-Pod" using data from when they were co-located together before and after the field deployment, and (3) apply the universal calibration model to the normalized sensor data from each Pod. The second U-Pod co-located with the reference instrument allows for validation of this method.

For the Los Angeles deployment, sensor-specific calibration models unique to each Y-Pod were used. As shown in Figure 5, the Y-Pods were all co-located together with reference instruments before and after the field deployment providing the opportunity to generate and use sensor-specific models. Additionally, one Y-Pod in Los Angeles was deployed with a reference instrument throughout the field deployment providing an additional set of validation data (referred to as Validation 2). This data offers the opportunity to calibrate the Los Angeles data using both sensor-specific calibration models and a universal calibration model – a direct comparison demonstrating the relative performance of these two methods. This offers





an informative comparison as there may be instances where only one method is possible given logistics, such as access to reference instruments. Another advantage to this universal calibration model approach would be that the calibration models are not extrapolating in time as the training data would cover the complete field deployment period.

## 3. Results & Discussion

### 3.1 Differences in Reference Data and Environmental Conditions that Impact Calibration

Different sampling environments necessitate the use of different strategies to produce the strongest calibration for each dataset. Reasons for this may be differences in local sources or metrological trends. Figures 6a and 6b illustrate the difference in temperature and humidity values observed during calibration verses validation periods for both locations. In Colorado, the temperature and absolute humidity observed during the validation period is generally well represented by the
data collected during the calibration period, although there are some high and low humidity values at certain temperatures that fall outside of the calibration parameter space. Conversely, in Los Angeles, the full range of temperature and humidity values observed during the validation period are captured in the calibration period. However, the Los Angeles data has many temperature/humidity combinations that are unique to the validation period.

Other sensor limitations must be considered as well, for example relatively slow sensor response. A low-cost sensor with an
operating principle relying on chemical reactions may not have time to fully detect a passing plume (Arfire et al., 2016) whereas this is not an issue for high-quality reference instruments that rely instead on optical properties. This limitation may result in sensor data that is fundamentally different from reference data further complicating the calibration model generation. One option for addressing this issue, explored below, is filtering short-duration reference data features prior to model generation. Demonstrating the need for this step, Figures 6d and 6e each show three days of data from the reference
monitors in which the diurnal patterns are similar, but the Colorado data also includes short-term enhancements or 'spikes' in methane possibly from the oil and natural gas extraction activity in the study region. The histogram in Figure 6c depicts the changes in methane values for each data set from minute to minute, further highlighting instances in the Colorado data where methane levels change by .5 or even 1 ppm over the course of a minute. These differences in the environmental parameter spaces emphasize the need to customize quantification methods to each dataset.

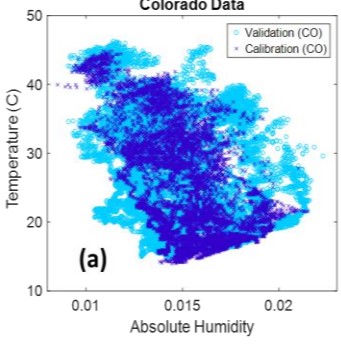
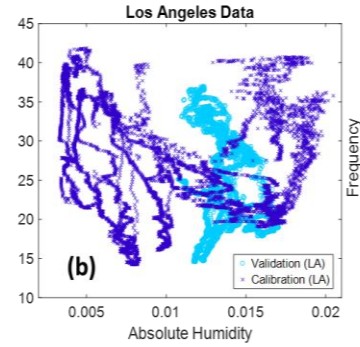
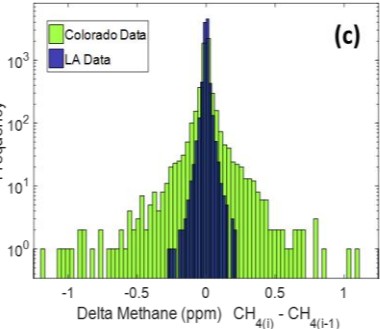
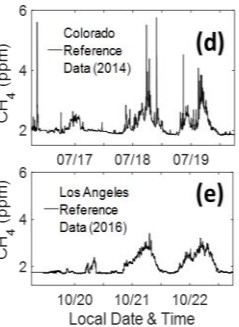



**Figure 6: Subplots 6a and 6b are the temperature and humidity values observed during calibration and validation periods in Colorado and Los Angeles respectively. While subplot 6c is a histogram of the changes in methane from minute to minute (for the reference data); and 6d and 6e are three days of minute-resolution data from the reference instruments in Colorado and Los Angeles respectively.**

## 3.2 Comparing Calibration Models

Table 3 contains the resulting statistics for each of the models described in Table 2 for one Colorado U-Pod and three Los Angeles Y-Pods. Three Y-Pods were selected randomly to facilitate analysis of the universal model and provide an initial indication of calibration model consistency across different sensors. This table lists the coefficient of determination ($R^2$) and the root mean squared error (RMSE) as well as the mean bias for the validation data. In all cases, these are the statistics for the fitted sensor data (converted into ppm $CH_4$) versus the reference methane data. Note, the second value in the Colorado data is the result when a filter is used to remove short-term spikes from the reference and sensor data. Filtering the Los Angles reference data did not change any of the statistics for that dataset and therefore was not used. This filter removes spikes that are greater than twice the past hour's standard deviation and last less than five minutes in duration.

**Table 3: Calibration Model Generation and Validation Results**

| # | Mdl | POD | CALIBRATION DATA | | VALIDATION DATA | | |
| | | | $R^2$ | RMSE (ppm) | $R^2$ | RMSE (ppm) | Mean Bias |
|---|---|---|---|---|---|---|---|
| 1 | Mdl1 | CO; Filt. | 0.463; 0.469 | 0.391; 0.369 | 0.456; 0.459 | 0.398; 0.378 | 0.014; 0.007 |
| | | LA 1 | 0.244 | 0.561 | 0.550 | 0.438 | -0.381 |
| | | LA 2 | 0.257 | 0.541 | 0.556 | 0.444 | -0.388 |
| | | LA 3 | 0.229 | 0.582 | 0.552 | 0.434 | -0.379 |
| 2 | Mdl3 | CO; Filt. | 0.498; 0.507 | 0.367; 0.346 | 0.329; 0.331 | 0.576; 0.550 | -0.035; -0.040 |
| | | LA 1 | 0.514 | 0.310 | 0.402 | 0.240 | 0.024 |
| | | LA 2 | 0.522 | 0.304 | 0.434 | 0.248 | -0.079 |
| | | LA 3 | 0.552 | 0.286 | 0.481 | 0.225 | 0.003 |
| 3 | Mdl4 | CO; Filt. | 0.574; 0.590 | 0.314; 0.292 | 0.392; 0.401 | 0.483; 0.453 | -0.038; -0.035 |
| | | LA 1 | 0.500 | 0.319 | 0.380 | 0.252 | 0.070 |
| | | LA 2 | 0.479 | 0.331 | 0.368 | 0.249 | 0.045 |
| | | LA 3 | 0.526 | 0.301 | 0.438 | 0.250 | 0.096 |
| 4 | Mdl4_1Int | CO; Filt. | 0.596; 0.625 | 0.300; 0.271 | 0.423; 0.449 | 0.437; 0.383 | -0.024; -0.011 |
| | | LA 1 | 0.747 | 0.184 | 0.518 | 0.224 | 0.047 |
| | | LA 2 | 0.765 | 0.176 | 0.558 | 0.212 | 0.026 |
| | | LA 3 | 0.752 | 0.181 | 0.527 | 0.225 | 0.069 |
| 5 | Mdl4_2Int | CO; Filt. | 0.588; 0.618 | 0.305; 0.275 | 0.432; 0.462 | 0.441; 0.384 | -0.019; -0.011 |
| | | LA 1 | 0.753 | 0.181 | 0.496 | 0.229 | 0.053 |
| | | LA 2 | 0.776 | 0.171 | 0.536 | 0.218 | 0.035 |
| | | LA 3 | 0.752 | 0.181 | 0.527 | 0.226 | 0.067 |
| 6 | Mdl4_1Int_Tr | CO; Filt. | 0.593; 0.622 | 0.302; 0.273 | 0.425; 0.451 | 0.440; 0.385 | -0.034; -0.019 |




| | | | | | | | |
|---|---|---|---|---|---|---|---|
| | | LA 1 | 0.737 | 0.189 | 0.512 | 0.341 | 0.264 |
| | | LA 2 | 0.761 | 0.177 | 0.457 | 0.389 | 0.316 |
| | | LA 3 | 0.745 | 0.184 | 0.525 | 0.380 | 0.316 |
| 7 | Mdl4_2Int_Tr | CO; Filt. | 0.588; 0.616 | 0.305; 0.275 | 0.440; 0.465 | 0.430; 0.379 | -0.041; -0.027 |
| | | LA1 | 0.784 | 0.167 | 0.627 | 0.198 | 0.061 |
| | | LA2 | 0.813 | 0.151 | 0.667 | 0.202 | 0.096 |
| | | LA3 | 0.776 | 0.169 | 0.655 | 0.217 | 0.120 |
| 8 | Mdl5_1Int | CO; Filt. | 0.597; 0.625 | 0.300; 0.271 | 0.426; 0.451 | 0.435; 0.382 | -0.021; -0.009 |
| | | LA 1 | 0.809 | 0.154 | 0.630 | 0.206 | 0.076 |
| | | LA 2 | 0.812 | 0.153 | 0.635 | 0.202 | 0.070 |
| | | LA 3 | 0.805 | 0.156 | 0.626 | 0.213 | 0.095 |
| 9 | Mdl5_2Int | CO; Filt. | 0.588; 0.618 | 0.305; 0.274 | 0.438; 0.466 | 0.437; 0.382 | -0.014; -0.008 |
| | | LA1 | 0.827 | 0.146 | 0.571 | 0.225 | 0.092 |
| | | LA2 | 0.833 | 0.142 | 0.580 | 0.222 | 0.091 |
| | | LA3 | 0.819 | 0.149 | 0.579 | 0.229 | 0.108 |
| 10 | Md5_1Int_Tr | CO; Filt. | 0.594; 0.622 | 0.302; 0.272 | 0.428; 0.453 | 0.437; 0.384 | -0.032; -0.018 |
| | | LA1 | 0.800 | 0.158 | 0.709 | 0.282 | 0.225 |
| | | LA2 | 0.807 | 0.154 | 0.630 | 0.341 | 0.285 |
| | | LA3 | 0.795 | 0.160 | 0.678 | 0.332 | 0.281 |
| 11 | Mdl5_2Int_Tr | CO; Filt. | 0.588; 0.617 | 0.305; 0.275 | 0.445; 0.467 | 0.426; 0.377 | -0.038; -0.026 |
| | | LA1 | 0.820 | 0.149 | 0.756 | 0.160 | 0.040 |
| | | LA2 | 0.831 | 0.143 | 0.734 | 0.180 | 0.083 |
| | | LA3 | 0.804 | 0.156 | 0.731 | 0.190 | 0.104 |

Figures 7a and 7b provide a graphical representation of the same statistics from Table 3 and emphasize the differences in results between the two datasets. For the Colorado data, the greatest improvement in fit was observed when time was added as a predictor, but then the results level off with no major improvements as the models increase in complexity. However, the results consistently returned a higher $R^2$ and lower RMSE when short-term methane spikes were filtered prior to model generation. In the Los Angeles dataset, there was continual improvement as models increased in complexity, with the most complex model producing a high $R^2$ and low RMSE as well as the most consistency across both the calibration and validations sets.



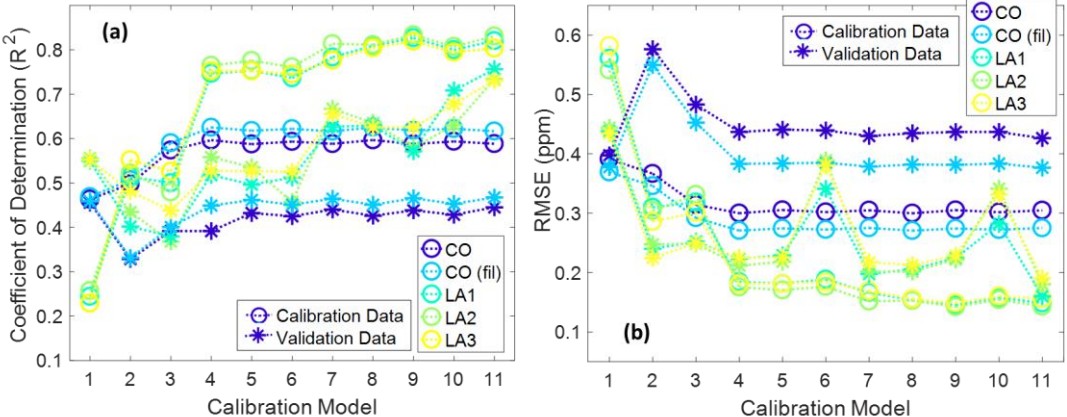

**Figure 7: Plotted R² (left, a) and RMSE (right, b) for all models, circle markers indicate results from calibration generation (using the training data) and asterisk markers indicate results from the application of the models to the validation data (or testing data).**

Figures 8 and 9 provide plots of the 'best-fitting' calibration models for each dataset based on regression statistics, consistency across calibration and validation data, and an analysis of the residuals. For the Colorado data, the selected model was the simplest well-fitting model, the fourth model. While for the Los Angeles data, the selected model was the most complex model tested, the eleventh model. With regards to both data sets, these models produced the most normal residuals, that also did not exhibit major trends with the predictors, and close one-to-one relationship between the fitted sensor data and the reference data. The time series plots also display the performance of the calibration model on the validation data set.

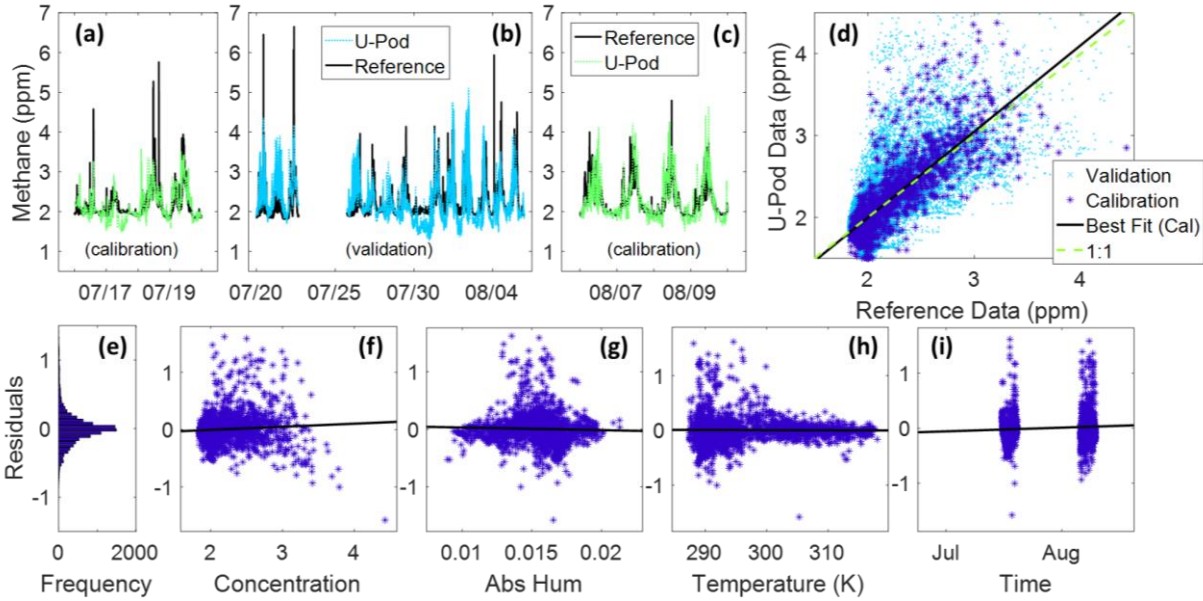

**Figure 8: 'Best-fitting' model for the Colorado data with residual analysis, model: MDL4_1INT (for validation data, RMSE = .383 ppm, mean absolute percent error = 12.13%). Subplots 8a, 8b, 8c are timeseries of the reference data and converted sensor data. Subplot 8d is a scatterplot of the same data. Subplots 8e, 8f, 8g, 8h, and 8i are the residuals from the claibration generation.**





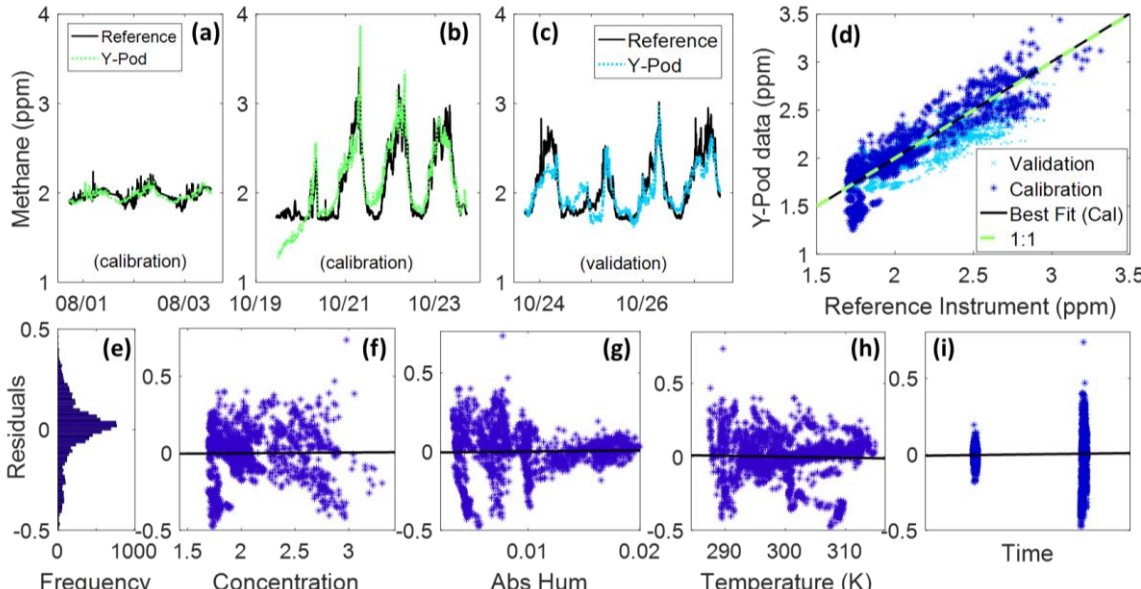

**Figure 9: 'Best-fitting' model for the Los Angeles data with residual analysis (LA1 shown), model: Mdl5_2Int_Tr (for validaiton data, RMSE = .160 ppm, mean absolute percent error = 5.75%). Subplots 9a, 9b, 9c are timeseries of the reference data ans converted sensor data. Subplot 9d is a scatterplot of the same data. Subplots 9e, 9f, 9g, 9h, and 9i are the residuals from the claibraiton generation.**

As demonstrated by these two datasets, calibration models are not "one-size-fits-all". While the deployments in Los Angles and Colorado occurred at roughly the same time of year, the best-fitting calibration models and regression results proved to be quite different. This speaks to the need to consider the environmental and pollutant parameter space both when planning a deployment and when processing data. For example, more complex temperature and humidity behavior may require more complex corrections. Additionally, if there is little overlap between conditions observed during calibration, validation, and field deployment then less dependable calibrations will result. Likely, there are factors beyond environmental parameter space driving differences between sensor and reference data. In that vein, it is important explore the operational differences between the reference instrument and the sensors, including distance apart and proximity to significant sources. Here, we compensated for those operational differences by filtering 'spikes' from the reference data; another modification could be to use a different averaging time such as hourly instead of minute data.

One feature of the models that applied to both datasets was a correction for sensor drift over time, emphasizing the importance of collecting data that either bookends or spans the duration of the field deployment. Even though the final models selected here differed, both included a correction for sensor drift over time and a pre or post only calibration would not have allowed for this correction. Finally, this analysis demonstrates the importance of exploring different models, transformations of variables, and treatments of the data to find the model that provides the strongest methane estimates.



### 3.3 Sensor-Specific vs. Universal Calibration Models

The additional validation data (Validation 2) collected during the field deployment in Los Angeles facilitates a comparison of the sensor-specific verses the universal calibration model approach. Several calibration models were generated using this additional co-located data (Figure 5), including the two models selected in the previous section as 'best-fitting'. These

5    models were then applied to normalized data from the other two Los Angeles Y-Pods included in the previous section. The raw sensor data from the Figaro TGS 2600 sensors was normalized using a simple linear regression (the $R^2$ values for these regressions were 0.989 and 0.999 respectively).

**Table 4: Calibration and Validation Results for the Universal Calibration Method, (\* - normalized Y-Pod data)**

|  |  | CALIBRATION DATA | | VALIDATION DATA | | |
|---|---|---|---|---|---|---|
|  |  | $R^2$ | RMSE | $R^2$ | RMSE | Mean Bias |
| MDL1 (1) | LA1 | 0.154 | 0.493 | 0.209 | 0.856 | -0.396 |
|  | LA2* |  |  | 0.223 | 0.835 | -0.382 |
|  | LA3* |  |  | 0.201 | 0.836 | -0.377 |
| MDL3 (2) | LA1 | 0.500 | 0.214 | 0.404 | 0.565 | -0.249 |
|  | LA2* |  |  | 0.438 | 0.581 | -0.292 |
|  | LA3* |  |  | 0.419 | 0.548 | -0.256 |
| MDL4 (3) | LA1 | 0.496 | 0.215 | 0.397 | 0.577 | -0.261 |
|  | LA2* |  |  | 0.434 | 0.589 | -0.304 |
|  | LA3* |  |  | 0.412 | 0.560 | -0.268 |
| MDL4_1INT (4) | LA1 | 0.463 | 0.230 | 0.372 | 0.622 | -0.283 |
|  | LA2* |  |  | 0.411 | 0.624 | -0.324 |
|  | LA3* |  |  | 0.387 | 0.606 | -0.288 |
| MDL5_2INT_TR (11) | LA1 | 0.477 | 0.223 | 0.486 | 0.334 | -0.098 |
|  | LA2* |  |  | 0.532 | 0.348 | -0.139 |
|  | LA3* |  |  | 0.529 | 0.336 | -0.129 |

Similar to the results from Section 3.2, the same model (MDL5_2INT_TR, mdl 11) emerges as the strongest for this particular dataset given that the validation statistics include the highest $R^2$ and lowest RMSE. An important note is that overall the results using this method are not as strong as the results seen using the sensor-specific models in the previous section. One reason for this may be that we are attempting to fit roughly six weeks, rather than four days for the calibration

15    model generation meaning the model is attempting to cover a larger environmental parameter space. This might also explain why the results for the 'best-fitting' model are better for the validation period, which is much shorter. In any case, this calibration model approach provides useful information regarding methane levels (e.g., diurnal trends), as is demonstrated by Figure 10, and this method can be used to convert the normalized signals from other sensors to a ppm value when logistics limit the potential for co-locating all sensors, whether due to time constraints or the limited availability of power/space at a





co-location site. As the logistics of the Colorado deployment did not allow for sensor-specific calibrations, the universal calibration model approach is used below in Section 3.5 to convert the Colorado field data from all the U-Pods.

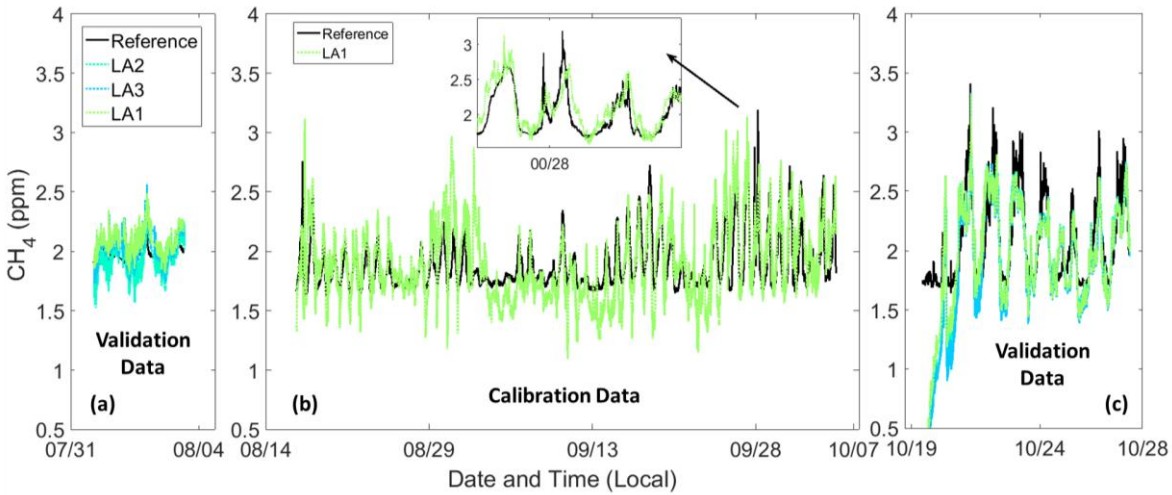

**Figure 10: Time series of the universal calibration model (for MDL5_2INT_TR) generated using the Validation 2 period (10b), and the application of the model to the pre and post-deployment colocation data (10a and 10c) for Y-Pods LA1 and the normalized data from LA2 and LA3.**

### 3.4 Further Sensor Quantification Considerations

Comprehensive best-practices to guide the use of low-cost air quality sensors have not been established. A recent workshop for low-cost sensors outlined some of the concerns shared throughout the research community including deployment logistics, data formatting and sharing, communication of uncertainty, etc. (Clements et al., 2017). With our datasets, we investigated three questions related to the development of best practices: the length of a co-location for a field normalization, additional dataset-specific filtering based on environmental parameters, and cross-sensitivities to non-methane pollutants.

### 3.4.1 Length of Co-location

Bootstrapping methods were applied to determine the variability and effectiveness of different co-location lengths for the Colorado data. A starting point in the complete dataset was randomly selected and consecutive data of varying lengths (0.5, 3, 7, or 14 days) was used to generate a calibration model. This model was then applied to the entire dataset for validation. For comparison purposes three different models were tested with 20 iterations for each model. The resulting statistics are plotted in Figure 11 along with error bars for +/- one standard deviation.





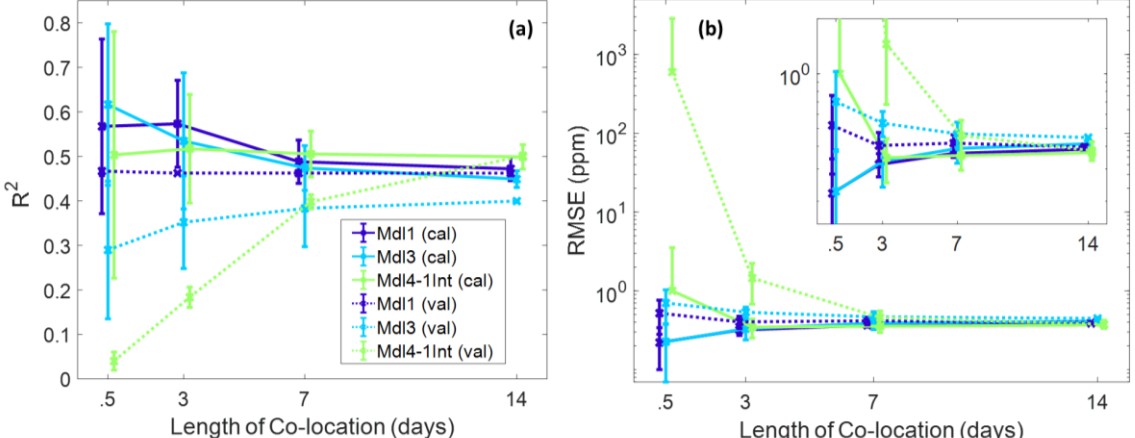

**Figure 11: $R^2$ (left, 11a) and RMSE (right, 11b) for the calibration model generation (based on a given length of time) and the application of those models to the complete Colorado data. Note, calibration data, or training data, selected using a random stating point in the complete Colorado dataset and the appropriate amount of consecutive data. The colors indicate the model, and solid lines indicate calibration results verses dashed lines indicating validation results.**

The simplest model (Mdl1), using only sensor signal and no environmental predictors, seems to perform consistently well for all lengths of time, however the residuals reveal strong trends with temperature and humidity indicating that these variables are not being corrected for. Given the analysis of the residuals, this model may provide useful information, but its implementation is also likely to be misleading. For example, this model may be useful in applications that do not require

detailed analysis or decision making based on the data, such as education and outreach in a K-12 classroom where sensors are used for labs or student projects (Collier et al., 2015). Taking into account residuals, Mdl3 provides some correction for temperature and humidity effects without overfitting on shorter co-location lengths. Mdl4_1Int, which includes time as a predictor, is the best performing model for co-location periods of two weeks. Given that time was a useful predictor in Section 3.2, the fact that the data is spanning two weeks is probably more important than having two full weeks of data. This

means that the co-location data must be long enough or span a long enough duration relative to the complete dataset in order to provide a time correction that does not lead to overfitting and poor performance on validation data. While greater complexity can provide a better calibration model, a sufficient amount of data must be used to avoid overfitting. Simply stated, the model selection should be appropriate to the data's characteristics and intended purpose.

### 3.4.2 The Impact of Model Extrapolation

The additional co-located data from Los Angeles (Validation 2) facilitates a more in-depth exploration of the outlier residuals and approaches that could improve the predictive power of the calibration model. For example, dataset specific filters were applied to remove values where extrapolation is likely occurring in field data. Extrapolation in this case, would be instances where one or more predictors are outside of the range of values used to train the calibration model. Table 5 provides the statistics that result from applying the calibration model with and without this added filtering. The unfiltered





dataset statistics are the same results explored in Section 3.2. All other statistics in the table were calculated after values not observed during calibration were removed. In the first filtered grouping in Table 5, instances where individual temperature or humidity values, etc. (primarily extreme values) not observed were removed. In the second grouping, all data combinations not observed during calibration were removed, meaning all instances where exact combinations of temperate,

5    humidity, etc. were not observed. The final filtering option, shown in the fourth section, applies knowledge of atmospheric composition to assist with filtering. In this instance, the atmospheric baseline of methane was used to filter out low sensor concentration values; the baseline was determined by the minimum value observed in the $CH_4$ reference data.

**Table 5: Additional Filtering to Improve Calibration Model Performance on Field Data**

| | Data | $R^2$ | RMSE | Mean Bias | n |
|---|---|---|---|---|---|
| Unfiltered datasets | Calibration | 0.820 | 0.149 | -0.002 | 9822 |
| | Validation 1 | 0.756 | 0.160 | 0.040 | 5461 |
| | Validation 2 | 0.527 | 0.166 | -0.032 | 71411 |
| Using only values represented in the calibration data | Temperature | 0.529 | 0.167 | 0.031 | 70340 |
| | Absolute Humidity | 0.529 | 0.166 | 0.031 | 71551 |
| | Resistance | 0.548 | 0.160 | 0.028 | 70851 |
| | Wind Speed/Wind Dir. | 0.526 | 0.167 | 0.031 | 69578 |
| | Temp, AH, & Res | 0.552 | 0.160 | 0.026 | 69473 |
| Using only paired values seen in the calibration data | Temp & AH | 0.564 | 0.157 | 0.030 | 51233 |
| | Temp, AH, & Res | 0.574 | 0.160 | 0.063 | 20059 |
| | Temp, Res, & WS/WD | 0.715 | 0.188 | 0.091 | 7645 |
| Adding atm. principles (A.P.) | $CH_4$ baseline applied | 0.552 | 0.160 | 0.047 | 71736 |
| Selected filtering | Temp, AH, Res, & A.P. | 0.581 | 0.153 | 0.041 | 69473 |

*Filtering Extrapolated Values* (vertical label on left side of table)

Additional filtering at nearly each stage yields some improvement in statistics, with the removal of the complete data combinations not seen during calibration resulting in the largest improvements, however this method also removes a large portion of the data. The combination of applied knowledge of atmospheric composition and the removal of extreme individual values not observed during calibration yields improvements while maintaining a substantial amount of the data.

15   This result, labeled 'Selected filtering', suggests that this more conservative version of filtering may be sufficient. Not only did this filtering result in a RMSE that is lower than the RMSE for first validation dataset (0.1525 ppm and 0.1601 ppm respectively), but also these improvements are visible in a plot of the data. Figure 12a provides an overview of the complete dataset and highlights where some of the under-predictions are corrected for before 9/7 and 9/17, likely driven by the filter utilizing atmospheric principles. Figure 12b, shows a close-up of a couple of days illustrating a reduction in over-prediction,

20   driven by the filter for either temperature, humidity, or resistance values.

As was demonstrated with the Validation 2 dataset, we expect that applying the same filtering to each deployed sensor's data should result in more reliable field data from all of the sites. Thus in addition to filtering data prior to calibration by removing short-duration enhancements in the reference data, filtering converted sensor data can improve the reliability of



calibration models. While the bounds for this type of filtering should be dataset-specific, this step could easily be an automated addition to low-cost sensor quantification procedures.

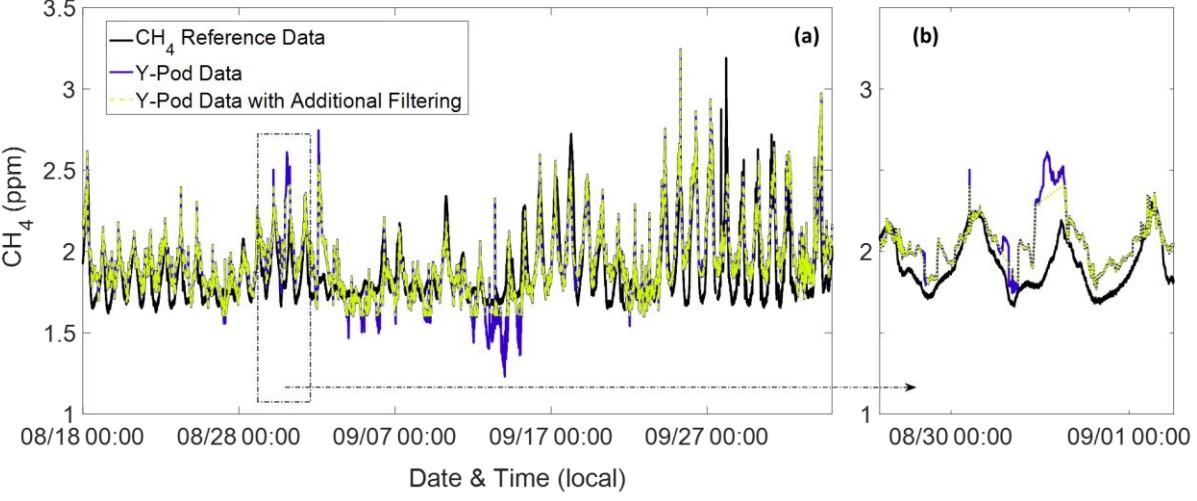

**Figure 12: 12a includes the complete Validation 2 dataset from the Los Angeles deployment. The statistics for each are: RMSE = .17 ppm and absolute percent relative error = 6.77% for the unfiltered data (blue); RMSE = .15 ppm and absolute percent relative error = 6.19% for the filtered data (green), with the reference data plotted in black. 12b is a close-up of approximately two days illustrating an instance where the filtering helped to reduce an over-prediction of methane concentrations.**

### 3.4.3 Sensor Cross-Sensitivities

Another common concern for low-cost sensors is cross-sensitivities to other gases. As discussed by Eugster and Kling (2012), the Figaro TGS 2600 sensor is reported to be sensitive to carbon monoxide as well as a few other hydrocarbons (Figaro, 2005b). This is not surprising, as each of these species can act as reducing gases at the sensor surface and therefore reduce the resistance to electron flow. While Eugster and Kling (2012) did not examine CO specifically given the absence of potential sources in their deployment area, they did perform an analysis of variance examining the effects of $CO_2$ and found no significant impacts. We applied the same analysis techniques to minute-resolution data to examine the effects of other gases, specifically CO and $O_3$. Given the information provided by the sensor manufacturer, we expected a cross-sensitivity to CO, but not to $O_3$; this analysis provided an opportunity to check these assumptions. Table 6 includes the resulting explained variance from each ANOVA, all of which included environmental parameters and time along with the following differences: Set 1 – $CH_4$ only, Set 2 – $CH_4$ and CO, Set 3 – $CH_4$ and $O_3$, Set 4 - CO only, and Set 5 - combined ($CH_4$ + CO) predictor.

**Table 6: Explained Variance from ANOVA Analyses on Figaro TGS 2600 Resistance Values ($R/R_0$) for Different Parameter Sets**

| Source of Variation | LOS ANGELES | | | | | COLORADO | | | | |
|---|---|---|---|---|---|---|---|---|---|---|
| | Set 1 | Set 2 | Set 3 | Set 4 | Set 5 | Set 1 | Set 2 | Set 3 | Set 4 | Set 5 |
| Temperature | 0.3% | 0.1% | 0.0% | 0.2% | 0.08% | 12.2% | 9.6% | 1.5% | 27.9% | 27.9% |



| | | | | | | | | | | |
|---|---|---|---|---|---|---|---|---|---|---|
| Abs. Humidity | 61.5% | 72.5% | 63.7% | 61.5% | 62.4% | 6.8% | 10.2% | 10.6% | 6.4% | 6.4% |
| Time | 0.0% | 0.0% | 0.1% | 0.0% | 0.0% | 8.3% | 8.9% | 10.8% | 3.7% | 3.7% |
| $CH_4$ | 16.8% | 2.6% | 14.2% | - | - | 29.2% | 21.8% | 20.5% | - | - |
| CO | - | 4.2% | - | 18.1% | - | - | 15.0% | - | 19.2% | - |
| $O_3$ | - | - | 0.3% | - | - | - | - | 2.7% | - | - |
| $CH_4 + CO$ | - | - | - | - | 19.8% | - | - | - | - | 19.2% |
| Residuals | 21.4% | 20.7% | 21.8% | 20.3% | 17.8% | 43.5% | 34.5% | 53.9% | 42.8% | 42.8% |
| Total | 100% | 100% | 100% | 100% | 100% | 100% | 100% | 100% | 100% | 100% |

The overall results varied between the deployments. For example, absolute humidity explained a high percentage of the variance in Los Angeles, while the temperature and humidity both played a role in the Colorado data. A commonality was that the sensor does exhibit a cross-sensitivity to CO, but not to $O_3$. In both cases, the inclusion of $O_3$ resulted in a higher

percentage of variance being attributed to the residuals, and the variance explained by the $O_3$ concentrations was 0.3% and 2.7% for Los Angeles and Colorado respectively. In contrast, the inclusion of CO in the ANOVA for the Colorado data resulted in a decrease of the variance explained by $CH_4$ from 29.2% to a still significant 21.8%, while 15.0% was explained by the new CO predictor. Notably, this set of parameters also resulted in the lowest portion of the variance being left to the residuals, suggesting that it provided the strongest set of among these five parameter sets. The inclusion of CO in the

ANOVA for the Los Angeles data yielded somewhat different results with the explained variance dropping drastically for $CH_4$ and being quite low for CO as well, at 2.6% and 4.2% respectively. This result is likely explained by the temporal correlation between the two gases obscuring the importance of each individually. The CO concentrations in Los Angeles were higher than those observed in Colorado and very well correlated with the $CH_4$ data as demonstrated in Figure 13. Further supporting this conclusion, parameter set 5 included a combined '$CH_4 + CO$' term and resulted in a higher portion of

the variance explained through this term at 19.8% verses $CH_4$ alone (16.8%) or CO alone (18.1%). This set also resulted in the lowest portion of variance left to the residuals. An additional sensitivity analysis that included speciated hydrocarbons was performed, that indicated that hydrocarbons also help to explain the Figaro TGS 2600 signal, but do not completely displace methane as a predictor (this analysis will be explored in-depth in a future paper). Thus, as demonstrated by our analysis the Figaro TGS 2600 sensor is cross-sensitive to carbon monoxide and likely to other hydrocarbons; the lack of

correlation between the Colorado CO and $CH_4$ allows us to examine the impacts of the CO cross-sensitivity more closely. Figure 14 shows a portion of the Colorado data with both reference and U-Pod methane plotted along with carbon monoxide data from a reference monitor. In Figure 14a spikes in CO correspond with overpredictions of methane (most notably on 8/8) and the scatterplot in Figure 14b highlights how overpredictions seem to coincide with elevated CO concentrations.



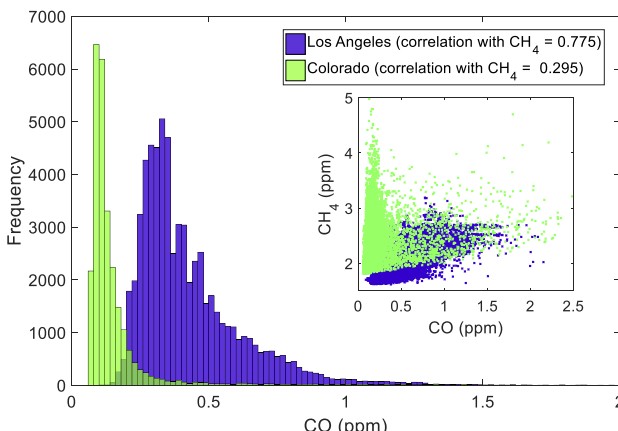

**Figure 13: Histogram of carbon monoxide data from the two deployments and scatter plot of carbon monoxide data vs. methane data also from the two deployments. Note, all data in these two plots are from reference instruments.**

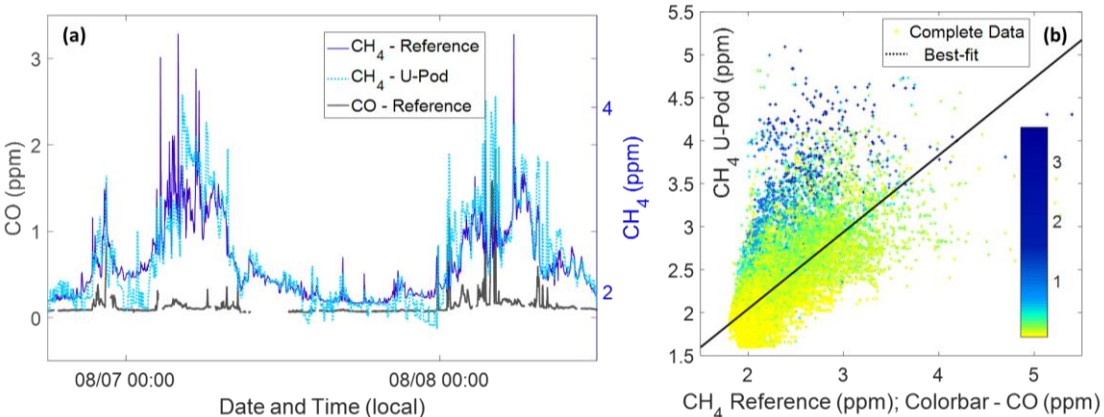

**Figure 14: Subplot 14.a depicts a timeseries of methane and carbon monoxide data from the reference monitors and converted U-Pod sensor data. Subplot 14.b depicts a scatterplot of reference methane data (x-axis) vs. U-Pod methane data (y-axis) with the points colored by carbon monoxide vales. Together they further show the Figaro TGS 2600's cross-sensitivity to carbon monoxide, illustrating how many over predictions correspond to instances when CO is at or above .5 ppm.**

Even though we sampled in more complex environments than previous deployments of this sensor (Eugster and Kling,

2012), we are still seeing a sizable proportion of the sensor data's variance explained by ambient methane concentrations. Although, these cross-sensitivities need to be addressed to discern which signals are driven by methane verses other pollutants, the Figaro TGS 2600 sensors are reacting in part to changes in ambient methane again providing useful methane estimates for applications where you need methane concentration resolution on the order of 0.2 – 0.4 ppm. Given the variety of low-cost sensors available, using the Figaro TGS 2600 sensors in the context of a sensor array could provide additional

signals at each deployment site resulting in more reliable data. Including multiple sensor signals with a neural network calibration approach may also improve the accuracy of the signal (Zimmerman et al., 2017; De Vito et al., 2008,





Huyberechts et al., 1997). Future analysis of the data collected in Los Angeles and continued use of this sensor in areas with complex mixtures will require carbon monoxide and NMHC impacts be considered.

**3.5 Ability to Assess Spatial Variability in the Northern Front Range of Colorado**

The universal calibration approach along with the 'best-fitting' calibration model (Figure 8) was used to convert the field data from the sensors deployed in Colorado. Following the same procedure outlined in Section 2.4 and examined in Section 3.3, the raw ADC values from each Figaro TGS 2600 sensor, from the post-calibration period, were normalized to the sensor signals in U-Pod P1 (the 'Main U-Pod') using sensor-specific simple linear fits. The calibration model was then applied to this normalized sensor data along with the temperature and humidity data from each U-Pod. An additional step was taken to detrend each set of converted sensor data by removing the best-fit linear trend from the whole dataset. It was necessary in this instance because the time correction incorporated in the calibration model appeared to be over- or under-correcting for different sensors. The choice to continue using this model was based on both the performance of the model observed in Section 3.2 and the fact that time appears to be a useful predictor for the Colorado data given the cross-sensitivity analysis in Section 3.4.3. Data from the pre-calibration period when six sensors were co-located with U-Pod P1 was used to verify that the application of this detrend function was appropriately correcting for the under or over correction of the model. One possible explanation for this difference in drift between the sensors may have been that five sensors were new and while the other ten (including the one in U-Pod P1) had been previously deployed. This difference in drifts was not observed in the Los Angeles data (Section 3.3), when all the sensors were new at the start of the deployment and were operated for the same amount of time throughout the deployment. The final step in preparing this data was to filter out data where the temperature and humidity values were outside of those ranges observed during calibration and to remove data where concentration values were lower than an expected minimum (atmospheric background) was observed; similar to the analysis in Section 3.4.2. In this case a conservative 1.6 ppm was used, roughly half of our RMSE below background methane levels. The largest amount of data removed from any U-Pod dataset as a result of this filtering was approximately 6%.

Table 7 presents statistics illustrating the correlation and RMSE for converted sensor data during co-location verses field deployment for both the two U-Pods continuously paired and the mean of all fifteen U-Pods. The result is high correlation when Pods are co-located and low correlation when U-Pods are deployed to their field sites. Highlighting that (1) there is consistency in the data provided through the universal calibration model and (2) that we are seeing quite a bit of variability across the field sites. Additionally, the RMSE for co-located U-Pods is continually less than the error we expect given the RMSE of .3832 ppm for the validation dataset.

**Table 7: Statistics for Colorado Data Converted Using the Universal Model Method**

|  | Post Co-location | | Deployed to Field Sites | |
| --- | --- | --- | --- | --- |
|  | R | RMSE (ppm) | R | RMSE (ppm) |
| Co-located Pair; P2 to P1 | 0.961 | 0.210 | 0.904 | 0.241 |
| Mean of all U-Pods to P1 | 0.866 | 0.180 | 0.025 | 0.556 |





This process provided minute-resolution methane estimates from 15 field sites, allowing for analysis of spatial data over different temporal scales. For example, Figure 15 includes roughly two days of data from four different deployment sites: P1 was our primary U-Pod located at the Platteville site, U-Pod E2 was located at the Boulder Atmospheric Observatory, U-Pod E3 was located at a water reclamation facility, and the U-Pod G1 was located at the Golden/NREL site. Even from this small

timeframe of data, we can see major differences between the sites. For example, there was a clear diurnal trend with elevated methane each night at the Platteville site. The high time resolution also allows us to observe short-term daytime increases at different sites, which were more sporadic and likely due to local emissions as there is typically more atmospheric mixing in the daytime (Bamburger et al., 2014). In contrast, the Golden/NREL site (U-Pod G1) exhibited relatively small variability in methane with differences between this site and the others well above our RMSE, suggesting significant differences in

methane concentrations between these sites. Figure 15b provides a reminder of where the oil and gas wells of the Wattenberg Field are in relation to these U-Pods. This high-resolution data (minute-median) allows for the study of individual emission events and possibly their correlation to nearby activity or regional trends.

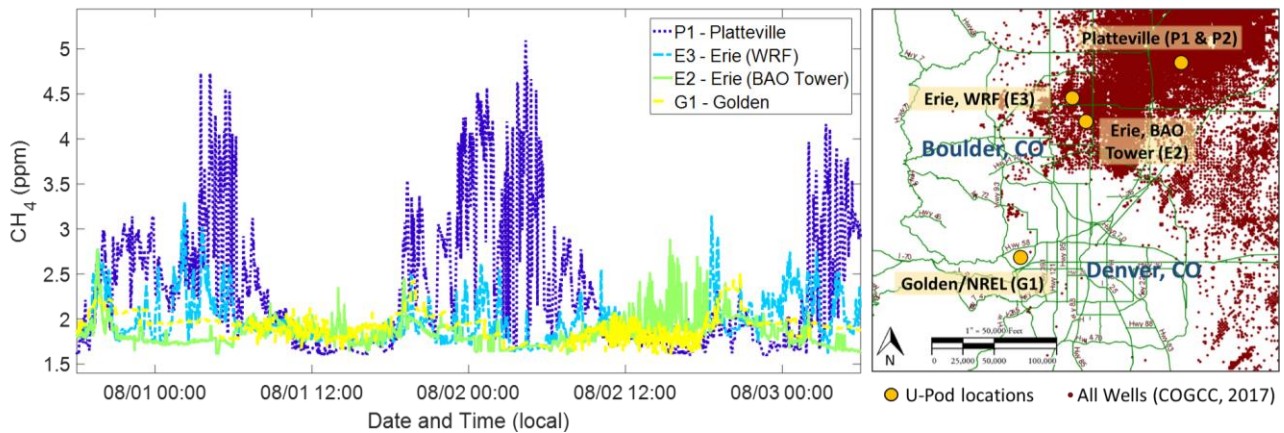

**Figure 15: Time series of minute-resolution data from four Colorado sites and a map showing the location of the four U-Pod with**
**respect to all active and inactive wells (COGCC, 2017).**

Figure 16 shows the day and night methane concentrations for each site throughout the deployment, grouped by region. This Figure also highlights the ambient background for methane +/- the U-Pod RMSE (0.3832 ppm) for this dataset on either side to illustrate that the enhancements above background were well beyond our expected error. The sensor in Golden/NREL (U-Pod G1) exhibited little variability across both daytime and night-time values, whereas all the sites in Erie, Frederick, and

Platteville exhibited larger ranges and larger night-time increases in methane likely contributed to by local or regional sources. At the majority of sites, over 50% of the data fell within the RMSE of typical background levels of methane. The middle 50% of the night-time data appears slightly shifted upward for U-Pods E11 and E3 in Erie (the two Pods located near the water reclamation facility). This trend was even more pronounced at the sites in Frederick and Platteville. Recall the well density show in Figures 2 and 15, illustrating no oil and gas activity around Golden, whereas we see higher density activity

in Erie and Frederick, with the highest density of activity around Platteville suggesting that one possible source driving this





elevated methane is emissions from oil and gas activity. We observe this trend at night when atmospheric mixing is more limited and the planetary boundary layer is lower.

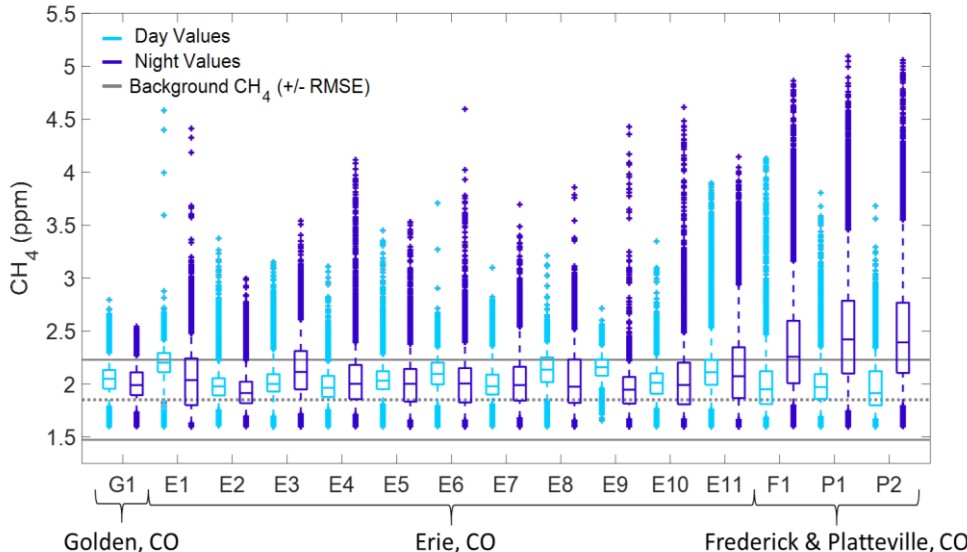

**Figure 16: Boxplots of all data for each U-Pod, grouped into daytime and night-time values, note – whiskers are the 5th and 95th percentile values. These U-Pods are then further grouped by region.**

Figure 17 further illustrates this point by showing the difference in 90th percentile values between the main U-Pod (P1) and all other U-Pods during the day (left) verses at night (right). The daytime differences are small, within +/- .2 ppm for all sites, possibly indicating effective daytime mixing. However, at night there is a clear gradient across the sites with little difference between the Pods in Platteville and increasing differences as we move to the edge of the gas field and outside of it, with a ~1 ppm difference for the Golden site and the site furthest west in Erie. As far as the trends go throughout the Erie sites, the two U-Pods in Erie furthest north, show the smallest difference with the Platteville Pods after the Fredrick Pod which was located much further into the gas field. These two Pods were also the ones located at the water reclamation facility and subject to an additional local methane sources. Interestingly though, the U-Pod furthest west in the Erie area was the only U-Pod in that grid located on the west side of the county line, which placed the Pod in Boulder County during a time when a moratorium beginning in 2012 was in place (Boulder County, 2017). This moratorium severely limiting new oil and gas development in the county. Although we cannot conclusively say this observation is the result of differing methane trends on either side of the county line, it is an indication of a question possibly worth revisiting using other data collected during the FRAPPE/DISCOVER-AQ campaigns. More importantly, this example demonstrates how low-cost sensors can offer preliminary or supplementary data that can help inform and guide future work.





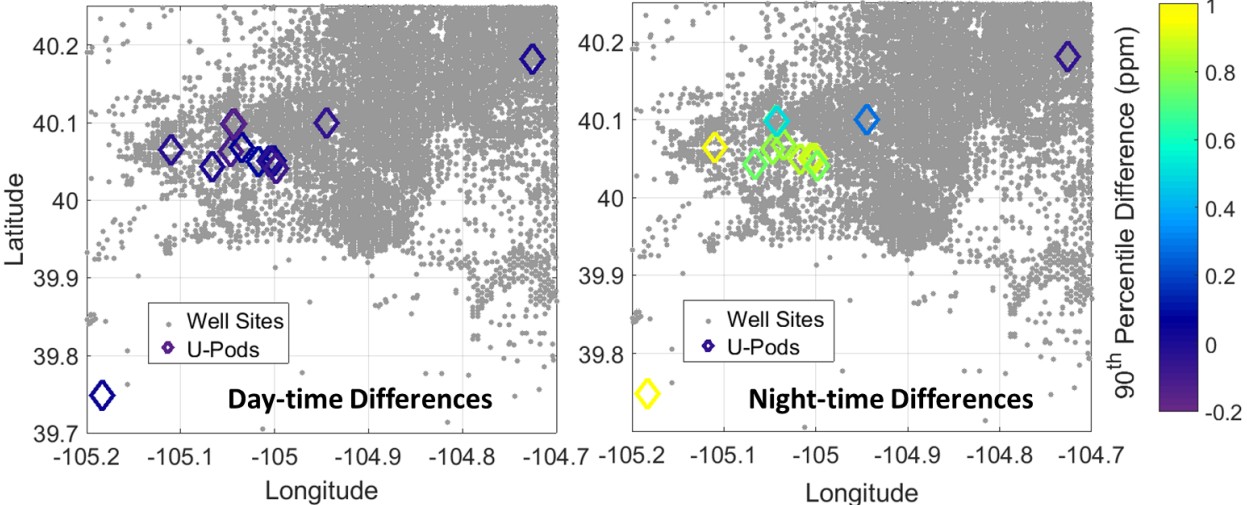

**Figure 17: Above well sites (COGCC, 2015) are plotted along with the differences between the 90th percentile value for the main Platteville U-Pod (P1) and each other U-Pod during the day (left) verses during the night (right). The colorbar indicates the magnitude of the difference in units of ppm CH₄, and the same colorbar scale is applied to both plots.**

Figure 18 provides another overview of the field data. In this plot, each hour of the day is averaged for each Pod using all available data – providing an indication of the diurnal patterns at each site. Again, we are seeing the night-time increase in methane occurring at the Platteville site and to a lesser extent an increase at the Frederick site. These increases continue to be well above background and the estimated measurement error, which supports the conclusion that night-time methane pooling was occurring in this location; a conclusion which is further supported by the observations of other researchers. Another

study also conducted during the FRAPPE/DISCOVER-AQ campaign found elevated levels of benzene at the Platteville Atmospheric Observatory, occurring primarily at night. These elevated concentrations were then attributed to local oil and gas activity, as opposed to another source like traffic, and the movement of planetary boundary layer (Halliday et al., 2016). Figure 18 indicates that something similar was occurring with methane at the same site, likely driven by one or more sources and the fluctuations of the planetary boundary layer fluctuations. Another study using data from 2013 found the mean level

of light alkanes in Platteville to be elevated 5-6 times above levels in Erie, and 9-15 times above levels in downtown Denver (Thompson et al., 2014). This trend of elevated alkanes in Platteville and lower levels in Erie also agrees with the gradients apparent in Figures 17 and 18 as we see the highest elevations in Platteville, moderate elevations in Frederick, and lower levels across our Erie sites. Overall, confirming the ability of low-cost sensors to provide unique information, in this case information regarding regional methane trends that was supported by studies that used conventional monitoring instruments

and sampling methods.



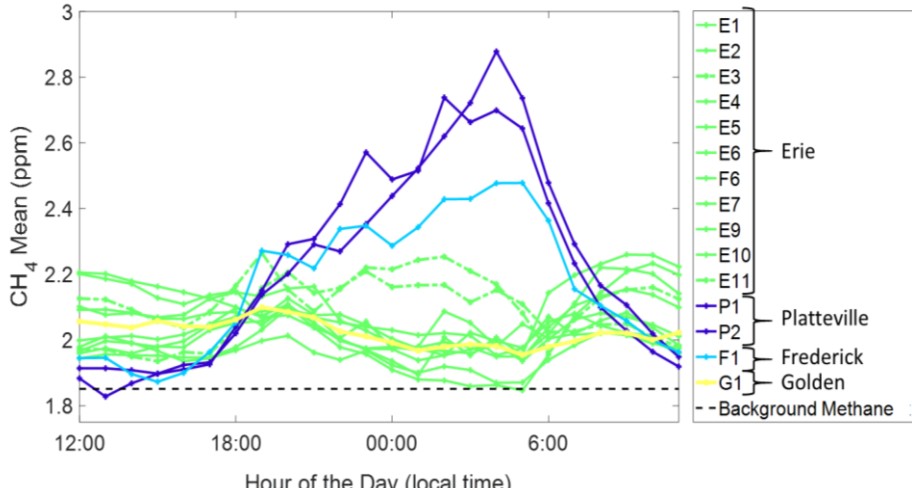

**Figure 18: Mean methane value for each hour of the day, for each U-Pod, grouped/colored by region.**

## 4. Conclusion

A common response to the question, "How good is low-cost sensor data?" is "it depends". It depends on what question you

are trying to answer, what data you intend to collect, how you would like to use the data, and what supporting measurements are available. As demonstrated by the quantification system applied to the two deployments examined in this paper, the use of low-cost sensors, certainly in the short-term, is likely to be heavily application-dependent and sensors should be calibrated and quantified to meet the needs of a given research question and in response to the conditions of a particular deployment. As low-cost sensor systems become easier to deploy and data processing becomes more automated, these systems have

tremendous potential. Their low-cost and portable nature allows for quick deployment across varied spatial scales, especially small, localized scales. Sensor data can already highlight potential "hotspots", which could lead to better allocation of resources or the detection of potential air quality issues sooner. When used in this context, the sensor system described herein can provide a useful estimate of methane concentrations that may serve as preliminary or supplementary data. In Los Angeles, we were able to provide a methane prediction despite the complexity of sources and this methane signal has the

potential to provide some insight into what is happening at the neighborhood-level, although special attention will need to be paid to possible confounders and cross-sensitivities. In Colorado, we were able to generate a dataset that can be examined on various temporal and spatial scales, as well as, data able to characterize regional trends that concur with the observations of other researchers. While more research into cross-sensitivities and other deployment issues is certainly necessary, this sensor system currently provides a powerful tool for understanding methane in communities near sources and additionally a tool

that is complementary to conventional monitoring methods.





**Data and Code Availability**

All sensor data (including final datasets and raw sensor data) and MatLab code used to process the data is available through the main author, please contact for access. All reference data used for analysis of the Colorado deployment was provided courtesy of the NATIVE Trailer team (Penn State University), and is available in the official DISCOVER-AQ database:

https://www-air.larc.nasa.gov/missions/discover-aq/discover-aq.html. Additional data for Los Angles analysis provided courtesy of the South Coast Air Quality Management District (note this data has not passed through the normal review process, and is therefore not QA'd and is unofficial data).

**Acknowledgments**

Funding provided through the MetaSense Project (NSF Grant CNS-1446912), the AirWaterGas Project (NSF-SRN CBET:

1240584), and through the DISCOVER-AQ Project (NASA). Thank you to all project partners during the DISCOVER-AQ/FRAPPE campaigns (NCAR, NOAA, CDPHE, US EPA), and all project partners in Los Angeles at the University of Southern California Keck School of Medicine, Redeemer Community Partnership (Nicole Wong), Sandy Navarro, William Flores, Esperanza Community Housing, and Occidental College. Additional thanks to all research and regulatory partners who assisted with site access and reference data: the NATIVE trailer team, Colorado Department of Public Health and the

Environment, and South Coast Air Quality Management District. Thanks to Christine Wiedinmyer (CIRES) for the deployment maps included in this paper (Section 2.2). Thanks to all monitor site hosts in Colorado and Los Angeles, and current and former members of the Hannigan Research Lab, especially Nicholas Masson and Drew Meyers for their work on the U-Pod/Y-Pod hardware and software, and Evan Coffey and Kira Sadighi for their assistance with the deployment in LA.

**Author Contribution**

ACO led the deployments in both locations with help from JC and HH in Colorado, and help from JJ in Los Angeles. ACO, MH, JC, RP, and JO all contributed to work quantifying low-cost sensors for methane. ACO organized this manuscript with assistance and feedback from all authors.

**Competing Interests**

The authors declare that they have no conflicts of interest.





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
