# Peer review of "Assessing a low-cost methane sensor quantification system for use in complex rural and urban environments"

_Atmospheric Measurement Techniques, 2017_

## Referee Comment (RC1) · Anonymous Referee #1 · 9 Feb 2018

This work presents a detailed analysis of the performance of one type of low cost metal-oxide sensor for methane detection. The study involved the deployment of multiple sensors both in the Colorado Front Range (for a period of ~1 month) and in urban Los Angeles (for ~2 months). These experiments were well designed within the constraints of larger studies, providing multiple opportunities for infield comparison of the sensors with established methane measurement technologies. The analysis presented provides a thorough comparison of several calibration strategies and possible explanations for the observed discrepancies. The authors show that the sensors are capable of providing useful information on spatial variability, whilst not overstating their capabilities. Overall I feel the paper is well written and a valuable contribution to the growing

body of work assessing the potential of low cost sensor technologies. I therefore recommend publication after the following minor comments have been addressed.

Minor comments:

1) It would be beneficial to the readers if the authors could provide more detail on how well the sensors agreed when they were co-located. Previous work (e.g. Jiao et al. 2016; Smith et al. 2017) has shown that variability between sensors can be significant. As Figs. 16 and 18 attribute observed differences between spatially distributed sensor signals to variations in methane concentrations at the different locations, an idea of the observed variability between co-located calibrated sensors would be useful. This could potentially be added as an extra panel to one or both of these figures?

2) As acknowledged by the authors, the analytical method used by the sensors in this study is sensitive to hydrocarbons other than just methane. As oil and gas operations co-emit a variety of hydrocarbons along with methane it is possible that the sensor response attributed to methane could instead be due to other hydrocarbons. Although the authors say that this will be covered in a future publication, I feel the issue does require some further discussion in this manuscript.

The authors state that the calibration parameters derived for the two study locations are significantly different, a problem well documented in the literature, but it would be useful to know which of the parameters in the equation show the most difference between the locations. Figure 6 shows that temperature and humidity observed in Los Angeles are within the range seen in Colorado, so one would expect the parameters associated with sensor response to these variables to be similar? It is likely that the hydrocarbons co-emitted with methane in Los Angeles and Colorado are different (geological basins often show characteristic hydrocarbon fingerprints). This would be visible as a difference in the parameters associated with the sensor sensitivity to methane if the sensor was responding more to the co-emitted hydrocarbons than the methane. The authors should comment further on this source of uncertainty.

3) I am not convinced by usefulness of the methane baseline filtering approach shown in Fig. 5 (A.P.). Removing data points that are below a known background methane would surely introduce a negative bias into the sensor data and unlike the other filters does not seem to be a test of the calibration model, but just a method of improving the agreement statistics. These signals below background levels could be indicative of other sensor dependencies not captured by the calibration model, (e.g. changing hydrocarbon mix (see comment 2) and thus contain useful information. I would suggest omitting this filter or providing more justification of why it is a valid approach.

Typographical errors:

Pg 15 line 12 "it is important explore" should read "it is important to explore" Pg 19 line 4 "temperate" should read temperature

References

Jiao W., et al.: Community Air Sensor Network (CAIRSENSE) project: evaluation of low-cost sensor performance in a suburban environment in the southeastern United States. Atmos. Meas. Tech., 9, 5281–5292, 2016.

Smith K. R., et al.: Clustering approaches to improve the performance of low cost air pollution sensors. Faraday Discuss., 200,621, 2017.

---

## Referee Comment (RC2) · Anonymous Referee #2 · 29 Mar 2018

Overall this paper provides an in depth analysis of the use of metal oxide sensors for sensing of Methane in this case as part of the FRAPPE/DISCOVER-AQ and California community project. Investigating the use of this particular type of "small sensor" in a relatively complex and changeable environment is a timely investigation. The deployment environments are atypical in their proximity to petrochemical extraction, transport or storage activities/facilities however this adds to rather than detracts from the utility of this manuscript.

The particular focus on calibration is well received. The in depth discussion on calibration models adds to the literature around these and other low cost technologies. This

reviewer feels that additional information regarding the field normalisation and other colocation activities would be useful. Particularly the practical setup used (shared inlets, enclosed chambers, flow cells or simply proximity etc. . .) and how it was replicated reliably. Additionally, information on inter-sensor differences in baseline, noise and response would be useful as well as a very short discussion of sensor selection (if undertaken). Further it could be informative to consider any hysteresis in these sensors from transient high concentration episodes (either of the detected species or of confounding species/parameters) which may or may not be seen in the data (irrespective or response time relative to the reference instruments). A short discussion on the response time of the sensor and of the instrument set up in its case (and any calibration issues that may result from this) would be welcomed.

Sensor drift over time is addressed and discussed however this reviewer feel it might be useful to show the variation in instrument drift pod to pod (possibly as a supplemental figure) and campaign to campaign to help investigation of sensor ageing relative to reference grade instruments and potentially effective lifetimes for the use of these sensors.

This reviewer thinks that a short discussion of cross sensitivities would be useful. Including with environmental parameters. Similarly, a discussion on potential differential response to other hydrocarbons would be useful. Potentially an associated point; This reviewer feels that the removal of reported values below the local background is not fully justified or explored currently as it could point to additional interferents or modes of behavior of the sensors. A further comment on the rational for removal of data spikes could also feed into this.

The nature of the deployments within wider deployment domains, particular within FRAPPE/DISCOVER-AQ, provided opportunities for cross comparison between techniques was well used by the research team and provide pointers to effective deployment methodologies.

[Figure]

Minor typographic errors: "In that vein, it is important explore the operational differences": Missing "to" between "important" and "explore".

---

## Author Comment (AC2) · 4 May 2018

COMMENT: "Overall this paper provides an in depth analysis of the use of metal oxide sensors for sensing of Methane in this case as part of the FRAPPE/DISCOVER-AQ and California community project. Investigating the use of this particular type of "small sensor" in a relatively complex and changeable environment is a timely investigation. The deployment environments are atypical in their proximity to petrochemical extraction, transport or storage activities/facilities however this adds to rather than detracts from the utility of this manuscript."

RESPONSE: The authors appreciate the reviewer's overview of their manuscript and

their helpful comments.

COMMENT: "The particular focus on calibration is well received. The in depth discussion on calibration models adds to the literature around these and other low-cost technologies. This reviewer feels that additional information regarding the field normalisation and other colocation activities would be useful. Particularly the practical setup used (shared inlets, enclosed chambers, flow cells or simply proximity etc. . .) and how it was replicated reliably."

RESPONSE: The authors agree that additional information on the practical setup is especially valuable in this case as the setup varied for each co-location. The following text was added to address this point.

New text added to Section 2.4: "The setup of Y-Pods for these co-locations was governed by limitations at the site. In Colorado, Y-Pods were mounted to the railing of the Native Trailer (approximately 1.5 m above the trailer roof), which housed the reference instruments. The inlets to the reference instruments were approximately 2.5 m above the roof of the trailer and roughly 2 m away from the Y-Pods. For the first co-location in Los Angeles, the reference instrument was housed in a trailer in an open field. As we were not able to place the Y-Pods on the roof of the trailer, they were placed .75-1.5 m off the ground on the side of the trailer where the inlet was mounted. In this case, the Y-Pods were roughly 6 meters below and 3 m to the side of the inlet. For the second co-location in Los Angeles, the reference instruments were housed inside of a building. In this case the Y-Pod was mounted to a railing roughly 2 m off the roof, which put the Y-Pod approximately 1 m below the inlet. However, the Y-Pod was also approximately 10 m away from the inlet, as this location was secure and out of the way of operations at the reference site. We would expect the variability between co-location setups to be most important for short-term spikes in CH4 that do not pass over the Y-Pod and inlet evenly. As discussed in Section 3.1, our co-location site in Colorado experienced the most short-term CH4 spikes, whereas the changes in CH4 concentration at the two LA sites were more gradual in nature."

COMMENT: "Additionally, information on inter-sensor differences in baseline, noise and response would be useful as well as a very short discussion of sensor selection (if undertaken)."

RESPONSE: The authors agree with this comment as well. To provide more information on inter-sensor differences for co-located sensors and to demonstrate how much this variability increases when sensors are deployed, Figures 1 and 2 here (Figure 19 and Table 9 in the revised manuscript), have been added to the appendix of the manuscript showing the differences between P1 and the sensors for each other Y-Pod. These figures demonstrate how the inter-sensor differences are sometimes driven by offsets (resulting in biased data) and are sometimes driven by differences in response (resulting in a slope greater or less than 1). However, in every case the RMSE between the sensor pair is less than our expected uncertainty for paired sensors. In addition to adding the figure and table to the appendix, the following text has been added to provide a discussion on the inter-sensor differences

New text added to Section 3.5: "The details of each individual sensor versus P1 is available in Figure 19 and Table 9, in Appendix A. These details demonstrate the extent of inter-sensor variability for co-located sensors and the increased variability for deployed sensors. While there is some variability among correlation coefficients, for nearly all sensors the periods of enhanced methane fall along the 1:1 line and most offsets occur at lower methane concentrations. Additionally, all RMSE's for co-located sensor fall below our expected uncertainty, while the RMSE's for deployed sensors is larger than this uncertainty (with the exception of the P1/P2 pair, which were co-located during the deployment)."

COMMENT: "Further it could be informative to consider any hysteresis in these sensors from transient high concentration episodes (either of the detected species or of confounding species/parameters) which may or may not be seen in the data (irrespective or response time relative to the reference instruments)."

RESPONSE: The authors found this to be an interesting question and tried to examine transient high concentration episodes for evidence of hysteresis. We began by selecting episodes where the range of the reference methane signal was greater than or equal to 1.5 ppm, the signal returned more or less to its starting position, and all of this occurred in the span of five minutes. Then the Matlab (a computing software utilized for this data processing and analysis) 'finddelay' function was utilized to look for consistent lags in the sensor data, possibly the result of hysteresis. This function can be used to identify delays between two columns of data as it examines the cross-correlation between two signals for different lags. This function was used to check for hysteresis by calculating lags for short, high-concentration spikes and lags for the sensor response and recovery periods separately. Given that this is minute-median data, the reported lags are in minutes. A limitation of this approach is that the function outputs the smallest absolute value for the observed lag. Thus, there may be individual events where there is a larger lag on either the response or the recovery period that we are missing, but we can be sure that this method would identify any consistent lags. Using this we found the following:

Finddelay(Sensor Data, Reference Data), for complete dataset = -1 Finddelay(Sensor Data, Reference Data), for selected spikes = 0 Finddelay(Sensor Data, Reference Data), for positive slope on spikes = 0 Finddelay(Sensor Data, Reference Data), for negative slope on spikes = 0

This seems to indicate that there is not a consistent lag either in the sensor response or the recovery associated with high concentration episodes. However, looking at individual high concentration events, we can find examples of a lag during the recovery possibly the result of hysteresis in the sensor (Figure 3 here, compares two spikes, one demonstrating this lag). Given the variation in the results, the authors believe that this issue could be examined more effectively in a lab using a controlled chamber. It is also worth noting that humidity is confirmed to impact both sensor response and recovery (Wang et al., 2010); sensors can also be poisoned (when compound bond

permanently to the sensor surface) limiting their response and ability to recover (Masson et al., 2015a). As there is much complexity around this issue and the fact that we are utilizing field data (meaning we cannot be certain of all the confounding factors affecting a sensor) we feel we cannot adequately address this question in the manuscript and have not added any additional text to the manuscript on this issue.

COMMENT: "A short discussion on the response time of the sensor and of the instrument set up in its case (and any calibration issues that may result from this) would be welcomed."

RESPONSE: Sensor response time versus instrument response time is an important issue to consider during field deployments. The authors have added specific information on reference instrument response time to expand the discussion currently in the paper noting the relatively slow sensor response time as compared to reference instruments.

New text added to Section 3.1, paragraph 2: "For example, the manufacture of the Picarro cavity ring-down spectrometer cites a gas response time under three seconds (Picarro, Inc., 2015), while Baseline Mocon cites a response time less than five seconds for the Series 900 Methane/Non-methane Hydrocarbon Analyzer (Mocon, Inc., 2017). Given these quick response times and the high flow rates used for sampling by the reference instruments we would not expect a lag on the part of the reference instrument. The sensor failing to reach steady state when exposed to a short, high concentration plume, as a result of slow sensor response, would be more of a concern for calibration."

COMMENT: "Sensor drift over time is addressed and discussed however this reviewer feel it might be useful to show the variation in instrument drift pod to pod (possibly as a supplemental figure) and campaign to campaign to help investigation of sensor ageing relative to reference grade instruments and potentially effective lifetimes for the use of these sensors."

RESPONSE: The authors have attempted to illustrate the drift from pod to pod by fitting the sensor data that using a calibration model that does not include time as a predictor (Mdl3, Table 2) and then fitting time to the signal, in an effort to quantify the linear drift. This was performed for LA1, LA2, and LA3 and Figures 4 and 5 (here) depict the linear drift over time for the co-located sensors data as well as the Mdl3 coefficients for the Y-Pod. These results illustrate the similarities in the coefficients for different predictors and the similarity of the linear drift values (0.009, 0.015, 0.011 ppm/week). Furthermore, this is similar to the values calculated by Eugster and Kling (2012) for the same sensor: 0.0156 and 0.0140. However, these drift values are below our uncertainty ($\sim$0.01 ppm/week over approximately 10 weeks results in $\sim$.1 ppm drift), which reduces our confidence in these estimates. Below is the text we have added noting our additional look into drift, while also noting its limitations.

While the authors do feel it would be valuable to compare pod to pod drift between campaigns, we are concerned that given the slightly different sensor platforms (U-Pod vs. Y-Pod), and different data treatment and processing (i.e., using model-specific calibration models for LA and universal calibration model for CO), the results could be misleading. You can also see from the added text that the authors have emphasized that drift and effective lifetimes are issues that will need to be addressed in future work.

New text added to Section 3.2: "To examine the consistency of this drift between sensors, we compared the linear drift remaining when the data from the three LA Y-Pods was converted to concentrations using Mdl3, which does not include time as a predictor. The results were drift values of 0.009, 0.015, 0.011 ppm/week for LA1, LA2, and LA3 respectively. While these numbers are similar to those reported by Eugster and Kling (2012), these values are also below our expected uncertainty making them unreliable. Given the differences in the deployments and their lengths, we have an initial idea of drift and its consistency sensor to sensor, however, a better understanding of drift as well as the effective lifetime of sensors will be important for future use of this sensor."

Eugster, W., & Kling, G. W.: Performance of a low-cost methane sensor for ambient concentration measurements in preliminary studies, Atmospheric Measurement Techniques, 5(8), 1925–1934. doi: 10.5194/amt-5-1925-2012, 2012.

COMMENT: "This reviewer thinks that a short discussion of cross sensitivities would be useful. Including with environmental parameters. Similarly, a discussion on potential differential response to other hydrocarbons would be useful."

RESPONSE: Acknowledgement and discussion of cross-sensitivities is especially important when discussing the quantification of metal-oxide sensors. While the authors feel that cross-sensitivities to environmental parameters have been explored in many other sensor studies (see the existing text copied from the manuscript below), they do agree that more information on cross-sensitivities to hydrocarbons would be useful. To address this point, we have added the results from another analysis of variance on the sensor signal that includes speciated hydrocarbons. This has been added as a table (Figure 6 here, Table 7 in the revised manuscript) and further discussion in the manuscript. As stated in the manuscript, the sensitivity and selectivity of sensors to different hydrocarbons will be explored more in depth in future work, but this analysis provides a preliminary indication that other hydrocarbons do help to explain the sensor signal, but do not negate the role of methane.

From Section 2.4: "Field normalizations were used to generate calibration models for the sensors. Field normalization provides one approach to correcting for the cross-sensitivities low-cost sensors tend to exhibit with respect to temperature, humidity, and other trace gases (Spinelle et al., 2015, 2017; Sadighi et al., 2017, Masson 2015a, 2015b; Wang et al., 2010)".

New text added to Section 3.4.3: "As previously stated, the FRAPPE/DISCOVER-AQ campaigns offered many opportunities for co-location with high-quality instruments and there was also a PTR-MS sited at the PAO, providing speciated VOC measurements (Halliday et al., 2016). Future work will provide a more in-depth analysis of VOC sensitivity and selectivity for different metal oxide sensors; however, we have included here a preliminary look at this cross-sensitivity to other hydrocarbons. Table 7 provides the results of another sensitivity analysis in which the explanatory power of a few speciated hydrocarbons is examined. For simplicity, one hydrocarbon from different correlated groups was selected (e.g., benzene was selected from the aromatic group, which exhibited high correlation among species). This analysis illustrates that VOCs (particularly acetaldehyde and benzene) do help to more fully explain the variance in the sensor signal, but they do not displace methane. This is most apparent for Parameter Sets 5 and 6, in which we see the variance explained by residuals increase slightly and the variance explained by temperature increase quite a bit as this factor compensates for the missing methane. When methane is added back in for Parameter Set 7, along with all three VOCs and CO, the variance explained by the residuals is at its lowest and the variance explained by methane is at 10.1% and higher than the percentages for the individual hydrocarbons. Thus, as demonstrated by our analysis the Figaro TGS 2600 sensor is cross-sensitive to carbon monoxide and hydrocarbons, along with methane."

COMMENT: "Potentially an associated point; This reviewer feels that the removal of reported values below the local background is not fully justified or explored currently as it could point to additional interferents or modes of behavior of the sensors. A further comment on the rational for removal of data spikes could also feed into this."

RESPONSE: The authors appreciate the reviewer's observations regarding the inclusion of filtering to remove values below background and have chosen to add further rational for its inclusion. The authors feel that this filtering is valuable to leave in as it highlights the need for discussions around data processing for different purposes. Data processing such as this may be a necessary part of using data from low-cost sensors as these anomalies are likely to occur. That being said, the authors agree that this filter might remove valuable information about sensor behaviour and response. Therefore, to address this comment, the authors have added further justification in the text high-

lighting the benefits and drawbacks of using this filtering – highlighting that this type of processing may be useful for sharing sensor data. We have also added another figure to the appendix (Figure 7 here, Figure 20 in the revised manuscript) illustrating that for every instance where these underestimations were removed by this filter, the dynamic range of methane for was less than our expected uncertainty (RMSE = .18). Therefore, this association seems to indicate that the underestimations are related to environmental factors, or possibly a limit of detection issue. This has also been pointed out in the text with additional discussion and a note to see the additional figure added to the appendix.

New text added to Section 3.4.2: "The final filtering approach, utilizing atmospheric composition, should only be applied to sensor data selectively. Removing improbable values from sensor data that fall below zero or a known baseline may be a useful or even necessary strategy in certain situations. In dealing with air quality data, there are examples of additional processing being used to reduce negative values (Hagler et al., 2011), and examples of guidelines to remove negative values below a given threshold (US EPA, 2016). For work with sensor data, if the focus of the analysis is to understand enhancements over background captured by sensors, then removing improbably low values can elucidate these results. If preliminary data is being shared with the public, then flagging and removing improbable values can reduce confusion. Given the challenges in sensor quantification, this second example in particular warrants consideration by those using sensors in partnership with communities and the public. However, it is also likely that these underestimations contain valuable information about sensor behaviour and sensitivity; removing these values will also introduce a negative bias to the data. Accordingly, when using this type of processing, researchers will need to be clear about why this approach is useful and valid for given situations. For this data set, every instance where underestimations are removed coincides with days having a dynamic range of methane less than the expected uncertainty for the sensor data, which indicates that these underestimations may be connected to a limit of detection issue. Figure 20 (Appendix B) demonstrates this association."

New Sources:

Hagler, G. SW., Yelverton T. LB., Vedantham, R., Hansen, A. DA., Turner, J. R.: Post-processing method to reduce noise while preserving high time resolution in aethalometer real-time black carbon data, Aerosol and Air Quality Research., 11(5), 539-546, doi: 10.4209/aaqr.2011.05.0055, 2011.

United States Environmental Protection Agency (US EPA): Technical Note ‐ Reporting Negative Values for Criteria Pollutant Gaseous Monitors to AQS, information available at: https://www.epa.gov/sites/production/files/2017-02/documents/negative_values_reporting_to_aqs_10_6_16.pdf (last access: April 2018), Oct. 2016.

COMMENT: Minor typographic errors: "In that vein, it is important explore the operational differences": Missing "to" between "important" and "explore".

RESPONSE: The authors have corrected this typographical error in the revised manuscript and have reviewed the entire manuscript once more for other errors.
* * *
[Figure]

[Figure]

**Fig. 1.**

|          | Co-located | | Deployed | |
|----------|--------|--------|--------|--------|
|          | R      | RMSE   | R      | RMSE   |
| **E1**   | 0.914  | 0.117  | -0.159 | 0.581  |
| **E2**   | 0.940  | 0.084  | -0.095 | 0.594  |
| **E3**   | 0.909  | 0.284  | 0.168  | 0.619  |
| **E4**   | 0.863  | 0.201  | 0.050  | 0.591  |
| **G1**   | 0.737  | 0.200  | -0.179 | 0.565  |
| **E5**   | 0.849  | 0.132  | 0.054  | 0.529  |
| **P2**   | 0.961  | 0.210  | 0.904  | 0.241  |
| **E6**   | 0.847  | 0.166  | -0.061 | 0.535  |
| **E7**   | 0.864  | 0.124  | -0.067 | 0.572  |
| **E8**   | 0.931  | 0.336  | -0.186 | 0.599  |
| **F1**   | 0.819  | 0.182  | 0.221  | 0.550  |
| **E9**   | 0.931  | 0.095  | -0.454 | 0.655  |
| **E10**  | 0.929  | 0.107  | 0.053  | 0.585  |
| **E11**  | 0.624  | 0.208  | 0.104  | 0.571  |
| **Average** | 0.866 | 0.175 | 0.025 | 0.556 |
| **Std Dev** | 0.092 | 0.073 | 0.306 | 0.096 |

**Fig. 2.**

[Figure]

**Fig. 3.**

[Figure]

**Fig. 4.**

|  | MDL3 Coefficients | | | |
|---|---|---|---|---|
|  | Intercept | Concentration | Temp. | Abs. Hum. |
| **B2** | 0.116 | -0.658 | 0.007 | -90.788 |
| **C9** | -0.619 | -0.647 | 0.009 | -75.707 |
| **D2** | 0.524 | -0.629 | 0.005 | -81.178 |

**Fig. 5.**

| Source of Variation | Set 1 | Set 2 | Set 3 | Set 4 | Set 5 | Set 6 | Set 7 |
|---|---|---|---|---|---|---|---|
| Temperature | 9.6% | 17.4% | 10.1% | 10.9% | 33.7% | 28.9% | 16.3% |
| Abs. Humidity | 10.2% | 11.7% | 14.2% | 7.5% | 6.2% | 9.0% | 12.6% |
| Time | 8.9% | 4.5% | 9.2% | 4.4% | 3.0% | 2.8% | 5.8% |
| $CH_4$ | 21.8% | 12.3% | 14.2% | 18.5% | - | - | 10.1% |
| CO | 15.0% | 13.6% | 14.4% | 19.7% | - | 9.8% | 11.4% |
| Acetaldehyde | - | 7.5% | - | - | 13.4% | 8.9% | 6.5% |
| Benzene | - | - | 4.6% | - | 8.4% | 6.4% | 4.0% |
| Methanol | - | - | - | 0.9% | 0.4% | 0.1% | 0.3% |
| Residuals | 34.5% | 33.0% | 33.4% | 38.2% | 34.9% | 34.1% | 33.2% |
| Total | 100.0% | 100.0% | 100.0% | 100.0% | 100.0% | 100.0% | 100.0% |

**Fig. 6.**

[Figure]

**Fig. 7.**

---

## Author Response (AR1)

Point-by-Point Response, List of Changes to Manuscript, and Marked-up Manuscript

**Reviewer Comment 1 & Author Response**

COMMENT: "This work presents a detailed analysis of the performance of one type of low cost metaloxide sensor for methane detection. The study involved the deployment of multiple sensors both in the Colorado Front Range (for a period of ~1 month) and in urban Los Angeles (for ~2 months). These experiments were well designed within the constraints of larger studies, providing multiple opportunities for infield comparison of the sensors with established methane measurement technologies. The analysis presented provides a thorough comparison of several calibration strategies and possible explanations for the observed discrepancies. The authors show that the sensors are capable of providing useful information on spatial variability, whilst not overstating their capabilities. Overall I feel the paper is well written and a valuable contribution to the growing body of work assessing the potential of low cost sensor technologies. I therefore recommend publication after the following minor comments have been addressed."

RESPONSE: The authors appreciate the overview provided by the reviewer and would like to thank the reviewer for their thoughtful comments.

**Minor Comments:**

COMMENT: "1) It would be beneficial to the readers if the authors could provide more detail on how well the sensors agreed when they were co-located. Previous work (e.g. Jiao et al. 2016; Smith et al. 2017) has shown that variability between sensors can be significant. As Figs. 16 and 18 attribute observed differences between spatially distributed sensor signals to variations in methane concentrations at the different locations, an idea of the observed variability between co-located calibrated sensors would be useful. This could potentially be added as an extra panel to one or both of these figures?"

RESPONSE: The authors agree that more detail regarding the inter-sensor variability would be a valuable addition to the manuscript. The attached figure and table (Figures 1 and 2 here, Figure 19 and Table 9 in the revised manuscript) have been added in an appendix to the manuscript, along with additional text noting where to find this figure and table as well as text discussing the inter-sensor variability. These additions have been made to Section 3.5 as they expand on the information already provided in Table 8 (Table 7, prior to manuscript revisions).

New text added to Section 3.5: "The details of each individual sensor versus P1 are available in Figure 19 and Table 9, in Appendix A. These details demonstrate the extent of inter-sensor variability for co-located sensors and the increase in variability for deployed sensors. While there is some variability among correlation coefficients, for nearly all sensors the periods of enhanced methane fall along the 1:1 line and most offsets occur at lower methane concentrations. Additionally, all RMSE's for co-located sensor fall below our expected uncertainty, while the RMSE's for deployed sensors is larger than this uncertainty (with the exception of the P1/P2 pair, which were co-located during the deployment)."

COMMENT: "2) As acknowledged by the authors, the analytical method used by the sensors in this study is sensitive to hydrocarbons other than just methane. As oil and gas operations co-emit a variety of hydrocarbons along with methane it is possible that the sensor response attributed to methane could instead be due to other hydrocarbons. Although the authors say that this will be covered in a future publication, I feel the issue does require some further discussion in this manuscript.

The authors state that the calibration parameters derived for the two study locations are significantly different, a problem well documented in the literature, but it would be useful to know which of the

parameters in the equation show the most difference between the locations. Figure 6 shows that temperature and humidity observed in Los Angeles are within the range seen in Colorado, so one would expect the parameters associated with sensor response to these variables to be similar? It is likely that the hydrocarbons co-emitted with methane in Los Angeles and Colorado are different (geological basins often show characteristic hydrocarbon fingerprints). This would be visible as a difference in the parameters associated with the sensor sensitivity to methane if the sensor was responding more to the coemitted hydrocarbons than the methane. The authors should comment further on this source of uncertainty."

RESPONSE: The authors agree that the difference in optimal calibration model parameters between the two locations is likely the result, at least in part, of different hydrocarbon mixtures. Not only are Los Angeles and Colorado likely to have different hydrocarbon fingerprints associated with the geologic basin, but also differences in traffic and other sources will like to affect the background hydrocarbon mixture as well. To address this, the authors have added an additional analysis of variance on the sensor signal that includes some hydrocarbon species. This analysis supports the point in the paper that other hydrocarbons do help to explain the sensor signal, but they do not displace methane. The text below was added to provide context for and discussion of this analysis, and the table has been added as Table 7 to the revised manuscript (attached as Figure 3 here). An additional note is that methane is generally present in the atmosphere at much higher levels than other hydrocarbons, such as ethane. Therefore, while this sensor is likely responding to other hydrocarbons, methane is likely a big driver of sensor response.

New text added to Section 3.4.3: "In addition to this observed cross-sensitivity to CO, we expect that other hydrocarbons may affect the sensor response as well. This would be an important consideration for measurements made in areas with oil and gas activity where the pollutant mixtures may be complex. At the PAO site there was also a proton-transfer-reaction quadrupole mass spectrometry (PTR-QMS) providing speciated VOC measurements (Halliday et al, 2016). Future work will provide a more in-depth analysis of VOC sensitivity and selectivity for the two MOx sensors we are using; however, we have included here a preliminary look at this cross-sensitivity to other hydrocarbons. Table 7 provides the results of another sensitivity analysis in which the explanatory power of a few speciated VOCs is examined. For simplicity, one VOC from different well-correlated groups was selected (e.g., benzene was selected out of the aromatic species). This analysis illustrates that VOCs (particularly acetaldehyde and benzene) do help to more fully explain the variance in the sensor signal, but they do not displace methane. This is most apparent for parameter sets 5 and 6, in which we see the variance explained by residuals increase slightly and the variance explained by temperature increase quite a bit as this factor compensates for the missing methane. When methane is added back in for parameter set 7, along with all three VOCs and CO, the variance explained by the residuals is at its lowest and the variance explained by methane is at 10.1%, higher than the percentages for the individual hydrocarbons. Thus, the Figaro TGS 2600 sensor seems to be cross-sensitive to carbon monoxide and some hydrocarbons; effects that should be considered or mitigated in future uses of this sensor to estimate methane."

COMMENT: "3) I am not convinced by usefulness of the methane baseline filtering approach shown in Fig. 5 (A.P.). Removing data points that are below a known background methane would surely introduce a negative bias into the sensor data and unlike the other filters does not seem to be a test of the calibration model, but just a method of improving the agreement statistics. These signals below background levels could be indicative of other sensor dependencies not captured by the calibration model, (e.g. changing hydrocarbon mix (see comment 2) and thus contain useful information. I would suggest omitting this filter or providing more justification of why it is a valid approach." RESPONSE: The authors appreciate the reviewer's observations regarding the inclusion of this baseline filtering and have chosen to add further justification for its inclusion. The authors feel that this filtering is valuable to leave in as it highlights the need for discussions around data processing for different purposes. Data processing such as this may be a necessary part of using data from low-cost sensors as these anomalies are likely to occur. That being said, the authors agree that this filter will likely introduce a negative bias and might remove valuable information about sensor behavior and response. Therefore, to address this comment, the authors have added further justification in the text highlighting the benefits and drawbacks of using this filtering – highlighting that this type of processing may be useful for sharing sensor data. We have also added another figure (Figure 4 here, Figure 20 in the revised manuscript) to the appendix illustrating that for every instance where these underestimations were removed by this filter, the dynamic range of methane for was less than our expected uncertainty (RMSE = .18). Therefore, this relationship does seem to indicate that the underestimations are related to environmental factors, or it might possibly be a limit of detection issue. This has also been pointed out in the text with additional discussion and a note to see the additional figure added to the appendix.

New text added to Section 3.4.2: "The final filtering approach, utilizing atmospheric composition, should only be applied to sensor data selectively. Removing improbable values from sensor data that fall below zero or a known baseline may be a useful or even necessary strategy in certain situations. In dealing with air quality data, there are examples of additional processing being used to reduce negative values (Hagler et al., 2011), and examples of guidelines to remove negative values below a given threshold (US EPA, 2016). For work with sensor data, if the focus of the analysis is to understand enhancements over background captured by sensors, then removing improbably low values can elucidate these results. If preliminary data is being shared with the public, then flagging and removing improbable values can reduce confusion. Given the challenges in sensor quantification, this second example in particular warrants consideration by those using sensors in partnership with communities and the public. However, it is also likely that these underestimations contain valuable information about sensor behavior and sensitivity; removing these values will also introduce a negative bias to the data. Accordingly, when using this type of processing, researchers will need to be clear about why this approach is useful and valid for given situations. For this data set, every instance where underestimations are removed coincides with days having a dynamic range of methane less than the expected uncertainty for the sensor data, which indicates that these underestimations may be connected to a limit of detection issue. Figure 20 (Appendix B) demonstrates this association."

New Sources:

Hagler, G. SW., Yelverton T. LB., Vedantham, R., Hansen, A. DA., Turner, J. R.: Post-processing method to reduce noise while preserving high time resolution in aethalometer real-time black carbon data, Aerosol and Air Quality Research., 11(5), 539-546, doi: 10.4209/aaqr.2011.05.0055, 2011.

United States Environmental Protection Agency (US EPA): Technical Note - Reporting Negative Values for Criteria Pollutant Gaseous Monitors to AQS, information available at: https://www.epa.gov/sites/production/files/2017-02/documents/negative\_values\_reporting\_to\_aqs\_10\_6\_16.pdf (last access: April 2018), Oct. 2016.

COMMENT: Typographical errors:

"Pg 15 line 12 "it is important explore" should read "it is important to explore" Pg 19 line 4

"temperate" should read temperature"

REPSPONSE: The authors have corrected this typographical error in the revised manuscript and have reviewed the entire manuscript once more for other errors.

Figure 1 (Figure 19 in revised manuscript)

**Figure 2 (Table 9 in revised manuscript)**

|           | Co-located |       | Deple  | oyed  |
|-----------|------------|-------|--------|-------|
|           | R          | RMSE  | R      | RMSE  |
| E1 | 0.914      | 0.117 | -0.159 | 0.581 |
| E2        | 0.940      | 0.084 | -0.095 | 0.594 |
| E3 | 0.909      | 0.284 | 0.168  | 0.619 |
| E4 | 0.863      | 0.201 | 0.050  | 0.591 |
| G1        | 0.737      | 0.200 | -0.179 | 0.565 |
| E5        | 0.849      | 0.132 | 0.054  | 0.529 |
| P2        | 0.961      | 0.210 | 0.904  | 0.241 |
| E6 | 0.847      | 0.166 | -0.061 | 0.535 |
| E7 | 0.864      | 0.124 | -0.067 | 0.572 |
| E8 | 0.931      | 0.336 | -0.186 | 0.599 |
| F1        | 0.819      | 0.182 | 0.221  | 0.550 |
| E9        | 0.931      | 0.095 | -0.454 | 0.655 |
| E10       | 0.929      | 0.107 | 0.053  | 0.585 |
| E11       | 0.624      | 0.208 | 0.104  | 0.571 |
| Average   | 0.866      | 0.175 | 0.025  | 0.556 |
| Std Dev   | 0.092      | 0.073 | 0.306  | 0.096 |

| Source of
Variation | Set 1  | Set 2  | Set 3  | Set 4  | Set 5  | Set 6  | Set 7  |
|------------------------|--------|--------|--------|--------|--------|--------|--------|
| Temperature            | 9.6%   | 17.4%  | 10.1%  | 10.9%  | 33.7%  | 28.9%  | 16.3%  |
| Abs. Humidity          | 10.2%  | 11.7%  | 14.2%  | 7.5%   | 6.2%   | 9.0%   | 12.6%  |
| Time                   | 8.9%   | 4.5%   | 9.2%   | 4.4%   | 3.0%   | 2.8%   | 5.8%   |
| $CH_4$                 | 21.8%  | 12.3%  | 14.2%  | 18.5%  | -      | -      | 10.1%  |
| CO                     | 15.0%  | 13.6%  | 14.4%  | 19.7%  | -      | 9.8%   | 11.4%  |
| Acetaldehyde           | -      | 7.5%   | -      | -      | 13.4%  | 8.9%   | 6.5%   |
| Benzene                | -      | -      | 4.6%   | -      | 8.4%   | 6.4%   | 4.0%   |
| Methanol               | -      | -      | -      | 0.9%   | 0.4%   | 0.1%   | 0.3%   |
| Residuals              | 34.5%  | 33.0%  | 33.4%  | 38.2%  | 34.9%  | 34.1%  | 33.2%  |
| Total                  | 100.0% | 100.0% | 100.0% | 100.0% | 100.0% | 100.0% | 100.0% |

Figure 3 (Table 7 in revised manuscript)

Figure 4 (Figure 20 in revised manuscript)